# RAD: Towards Trustworthy Retrieval-Augmented Multi-modal Clinical Diagnosis

**Haolin Li**[1,2*] **Tianjie Dai**[3*] **Zhe Chen**[2,3] **Siyuan Du**[1,2]
**Jiangchao Yao**[2,3†] **Ya Zhang**[2,4,5] **Yanfeng Wang**[2,4†]

[1]College of Computer Science and Artificial Intelligence, Fudan University
[2]Shanghai AI Laboratory    [3]CMIC, Shanghai Jiao Tong University
[4]School of Artificial Intelligence, Shanghai Jiao Tong University
[5]Institute of Artificial Intelligence for Medicine, Shanghai Jiao Tong University School of Medicine

## Abstract

Clinical diagnosis is a highly specialized discipline requiring both domain expertise and strict adherence to rigorous guidelines. While current AI-driven medical research predominantly focuses on knowledge graphs or natural text pretraining paradigms to incorporate medical knowledge, these approaches primarily rely on implicitly encoded knowledge within model parameters, neglecting task-specific knowledge required by diverse downstream tasks. To address this limitation, we propose **R**etrieval-**A**ugmented **D**iagnosis (RAD), a novel framework that explicitly injects external knowledge into multimodal models directly on downstream tasks. Specifically, RAD operates through three key mechanisms: retrieval and refinement of disease-centered knowledge from multiple medical sources, a guideline-enhanced contrastive loss that constrains the latent distance between multi-modal features and guideline knowledge, and the dual transformer decoder that employs guidelines as queries to steer cross-modal fusion, aligning the models with clinical diagnostic workflows from guideline acquisition to feature extraction and decision-making. Moreover, recognizing the lack of quantitative evaluation of interpretability for multimodal diagnostic models, we introduce a set of criteria to assess the interpretability from both image and text perspectives. Extensive evaluations across four datasets with different anatomies demonstrate RAD's generalizability, achieving state-of-the-art performance. Furthermore, RAD enables the model to concentrate more precisely on abnormal regions and critical indicators, ensuring evidence-based, trustworthy diagnosis. Our code is available at this repository.

## 1 Introduction

The rapid development of multimodal learning [37, 46] has revolutionized numerous fields by enabling models to process and integrate diverse data types, including images, texts, audio, and structured records [4, 11, 77]. Biomedical applications particularly benefit from these advancements, given that diagnostic workflows inherently depend on multimodal evidence, ranging from radiographic imaging and reports to electronic health records (EHR) [16, 52, 75]. For instance, radiologists integrate X-ray or MRI scans with textual pathology reports, while clinicians combine electronic health records, vital signs, and even genomic data to form comprehensive patient profiles. Accordingly, recent research efforts have increasingly focused on developing multimodal architectures tailored to healthcare challenges, seeking to enhance diagnostic precision through cross-modal synergy [6, 32, 61, 68]. While these approaches demonstrate significant progress in integrating data from different modalities, they often overlook the foundational principles governing clinical decision-making.

---

*Equal Contribution.  † Correspondence to Jiangchao Yao (Sunarker@sjtu.edu.cn) and Yanfeng Wang (wangyanfeng622@sjtu.edu.cn).

39th Conference on Neural Information Processing Systems (NeurIPS 2025).

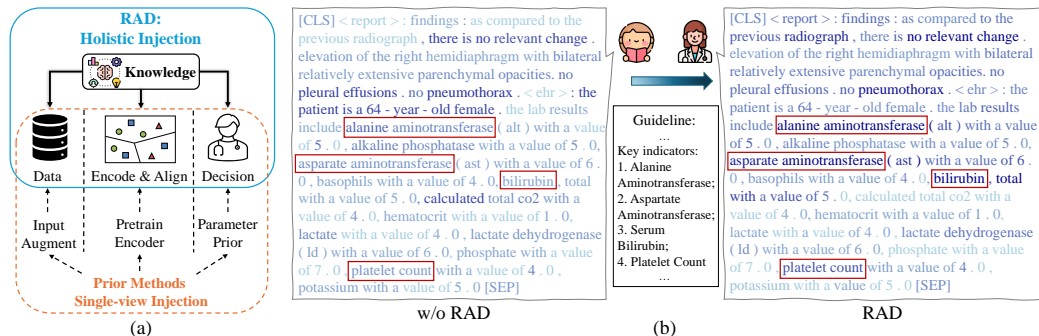

Figure 1: The design motivation of RAD. **Left**: Previous methods mostly focus on enhancing a single aspect of the diagnostic process, whereas our approach is holistic. **Right**: Visualization of model attention to textual content. Color intensity reflects attention magnitude, with red highlighting disease-critical indicators mentioned in the guideline. Models without explicit knowledge guidance exhibit limited focus on key indicators, whereas our model can make evidence-based diagnoses.

Medical analysis fundamentally differs from natural scene understanding in its strict adherence to evidence-based principles, relying heavily on structured protocols [31, 50]. Clinical decisions must be grounded in standardized diagnostic criteria derived from patient-specific symptoms, imaging findings, and laboratory results. This inherent rigor poses a critical challenge for black-box neural networks, whose vague decision-making mechanisms hinder trustworthy and practical deployment in clinical settings [19, 49, 53]. Consequently, there has been growing interest in integrating medical knowledge into AI models to simultaneously improve model performance and interpretability [8, 9, 73].

Existing approaches primarily focus on knowledge injection during pretraining phases. Researchers enhance the text encoders by pretraining them on large-scale medical corpora [30, 47] or leveraging structured knowledge graphs to imbue models with semantic relationships between biomedical entities [36, 48]. While effective in expanding the semantic coverage of text encoders, these approaches often struggle to explicitly integrate fine-grained knowledge tailored for downstream diagnostic tasks. To this end, we argue that effective knowledge integration requires task-centric, holistic alignment with disease-level knowledge throughout the entire diagnostic pipeline. As illustrated in Figure 1(a), our framework systematically integrates refined knowledge to guide input augmentation, feature extraction, and modality fusion, contrasting with prior methods confined to a single perspective. Figure 1(b) presents a case of the model's attention distribution over the input text. The previous model fails to concentrate on critical indicators, but focuses on obvious disease terms in the reports. While the RAD model can not only attend to these terms but also consider other guideline-recommended key indicators. The explicit knowledge guidance enables RAD to prioritize critical indicators tailored for the current disease, making trustworthy diagnoses aligned with clinical standards.

In this paper, we propose a holistic knowledge-injection framework RAD, which operates through three synergistic components spanning the entire diagnostic workflow. RAD begins by retrieving and refining disease-specific guidelines from diverse sources, flexibly adapting to downstream tasks in different scenarios. We then employ two modality encoders coupled with guideline-enhanced contrastive loss that explicitly aligns the modality-specific feature with the corresponding disease-guideline prototypes in the joint latent space. A dual decoder network is further developed to steer the cross-modal fusion process, which simultaneously incorporates disease labels and their corresponding guidelines to interact with fused multimodal features for final predictions. Through this systematic knowledge infusion paradigm, our framework achieves performance gains while establishing a traceable decision pathway grounded in clinical guidelines—a critical step toward clinically actionable AI. We further establish an evaluation system for model interpretability, which quantitatively assesses the model's adherence to guidelines through both textual indicators and visual localization. Combined with qualitative visualization, this system provides measurable evidence that RAD's decisions are driven by the injected knowledge. In summary, our contributions are three-fold:

- We propose RAD to systematically inject external medical knowledge into multimodal diagnosis models. RAD incorporates a guideline-enhanced loss and a dual-decoder structure to explicitly steer multimodal feature extraction and cross-modal fusion with disease-guideline prototypes.

- A dual-axis evaluation system for the interpretability of diagnosis models is developed, formulating both textual and visual metrics. This system enables quantitative analyses of the model's adherence to clinical guidelines, demonstrating the transparency and explainability brought by RAD.

- We aligned MIMIC-CXR [28] and MIMIC-IV [29] to construct the MIMIC-ICD53 dataset, covering three modalities with 53 types of disease. Extensive experiments on our dataset and three other public datasets demonstrate the superiority of RAD over SOTA baselines across various metrics.

## 2 Related Work

### 2.1 Multimodal Learning in Medicine

Recent years have witnessed significant advancements in the field of multimodal learning, with models such as CLIP [46], BLIP [33], and LLaVA [37] exhibiting remarkable capabilities in natural domains. These developments have spurred increasing interest in extending multimodal frameworks to the medical field, where the integration of diverse data modalities demonstrates prominent potential in diagnostic tasks. Current research focus lies in **multimodal pretraining** methods, which focus on cross-modal alignment between imaging and textual data to improve the representation transferability [7, 17]. ConVIRT [72] and GLoRIA [26] pioneered the application of CLIP-style architectures in the medical domain by constructing image-text pairs from radiology datasets. MedCLIP [56] and BiomedCLIP [70] addressed the scarcity of paired medical image-text data by leveraging multi-source datasets, achieving state-of-the-art performance. Beyond pretraining methods, **multimodal fusion** approaches aim at integrating information from different modalities for diagnostic applications [20, 57, 66]. MedFuse [21] introduced an LSTM-based temporal fusion method of time-series data and X-ray images. HEALNet [23] proposed a hybrid early-fusion method to learn from data sources with different structures. While these works have made significant strides, they often operate without explicit guidance from medical knowledge when addressing specific diagnosis tasks. In contrast, considering the evidence-based nature of medicine [44, 51], RAD explicitly incorporates task-specific knowledge to guide both representation extraction and multimodal fusion processes.

### 2.2 Medical Knowledge Injection

Injecting professional knowledge into AI models is a prevalent strategy to improve their domain-specific capabilities [40, 65]. Various techniques have been investigated to incorporate medical knowledge into the models. **Pretraining-based** approaches train the text encoder on extensive medical domain corpora, such as PubMedBERT [18] and HUATUO-GPT [69]. Other methods like KAD [71] and DRAGON [67] leverage structured knowledge graphs of medical entities for pre-training to enhance the text encoder's comprehension of medical terminology. While showing empirical effectiveness, these knowledge integration methods remain primarily confined to the pre-training phase, providing only implicit guidance during the subsequent diagnostic stage. With the rapid development of large language models (LLMs) [1, 35, 54], various **Retrieval-Augmented Generation** (RAG) methods have been proposed to enhance the generation process of medical LLMs [13, 61]. These methods dynamically retrieve external medical knowledge to improve the performance of LLMs on question-answering (QA) tasks [34, 60]. Building upon this foundation, multimodal RAG approaches further retrieve similar data samples (e.g., image-report pairs) for visual question-answering (VQA) [59, 74]. In contrast to RAG methods that *online* retrieve knowledge to augment input for *generative* QA/VQA tasks, our framework adopts a structured approach for *discriminative* tasks by performing *offline* retrieval of disease-specific knowledge, which is systematically incorporated to guide model training.

## 3 Method

In this section, we first present the problem formulation, followed by the detailed introduction of our proposed method, Retrieval-Augmented Diagnosis (RAD), which consists of guideline retrieval and refinement, guideline-enhanced feature constraint, and dual diagnostic network. Finally, we introduce our interpretability evaluation system. The overall framework of RAD is illustrated in Figure 2.

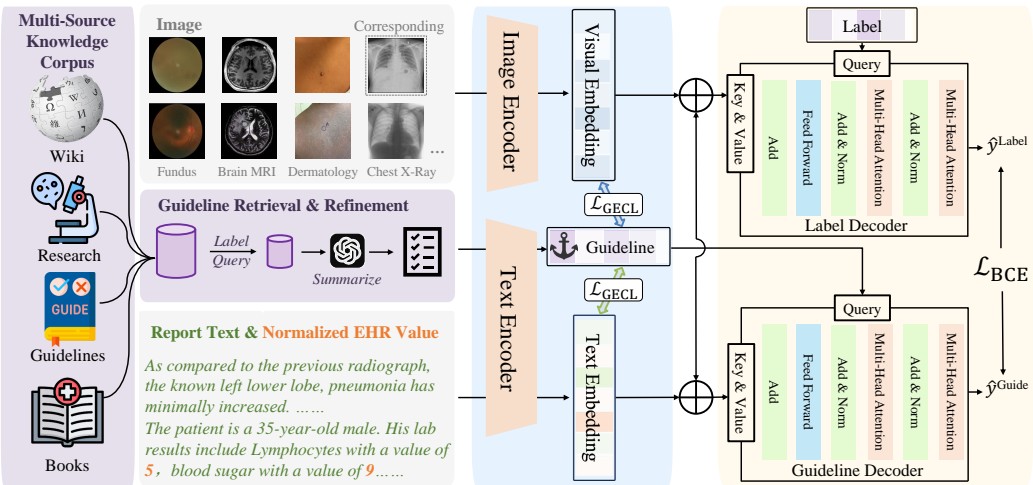

Figure 2: The overview of our Retrieval-Augmented Diagnosis framework, including multi-source medical knowledge retrieval and refinement, multimodal representation learning under the guideline constraint, and the dual diagnosis network. $\oplus$ represents the concatenation operation.

## 3.1 Problem Formulation

Given a training set of $N$ samples, $\mathcal{D} = \{(x_i, t_i, y_i)\}_{i=1}^{N}$, where $x_i$ represents a radiology image, $t_i$ is the report and electronic health records, and $y_i \in \{0, 1\}^m$ is the corresponding multi-label vector indicating the presence of $m$ diseases. The multi-source medical knowledge corpus is denoted as $P = \{p_i \mid i = 1, 2, \ldots, s\}$, where $p_i$ denotes the $i$-th source and $s$ denotes the number of sources. The guideline corresponding to the disease label is derived from multi-source retrieval and refinement. We denote this guideline as $g = \{g_i \mid i = 1, 2, \ldots, m\}$. The objective is to develop a multimodal model trained on $\mathcal{D}$, capable of accurately predicting the disease for any given multimodal sample.

## 3.2 Retrieval-Augmented Diagnosis

### 3.2.1 Guideline Retrieval and Refinement

**Knowledge-corpus.** To retrieve disease-related diagnostic knowledge, we collect medical knowledge from four distinct sources: "Wiki", "Research", "Guideline", and "Book". **Wiki** provides comprehensive descriptions of target diseases, such as formal medical definitions, and clinically relevant subcategories. **Research** incorporates the latest research articles from PubMed (a premier database of biomedical literature). These articles provide cutting-edge findings in disease mechanisms, diagnostic criteria, and therapeutic interventions. **Guideline** includes 45K clinical practice guidelines from 13 sources, providing rigorously vetted diagnostic criteria and treatment protocols for medical practitioners. **Book** consists of diverse medical textbooks, covering basic medical knowledge in surgery, medical imaging, and drugs, etc. More details of the corpus can be found in Appendix B.1.

**Disease Knowledge Retrieval.** For a given dataset with $m$ diseases, our objective is to retrieve the most relevant knowledge from the knowledge corpus $P$, including but not limited to: associated symptoms, imaging characteristics, and critical examination/laboratory indicators. We adopt Med-CPT [27], a dual-encoder model optimized for medical scenarios, as the retriever. Specifically, the article encoder $R_A(\cdot)$ is utilized to convert the corpus $P$ into dense vectors for retrieval. The disease names $E$ are used as the input query of the query encoder $R_Q(\cdot)$. The obtained embeddings are then used to calculate the similarity as $\text{Sim}(E, P) = R_Q(E)^\top R_A(P)$. For each disease with a name $e_i \in E$, we preserve the top-$k$ retrieved documents as:

$$\mathcal{C}_i = \underset{p_j \in P}{\text{Top-}k} \ \text{Sim}(e_i, p_j). \tag{1}$$

**LLM Refinement.** Given that retrieved documents $\mathcal{C}_i$ may contain content irrelevant to the diagnosis of the current disease and exhibit cross-source redundancy, directly combining the retrieved documents as the final guideline is suboptimal. In addition, the total document length often exceeds

the context window of the diagnosis model. To address these challenges, we employ large language models (LLMs) to perform automated summarization and refinement of $\mathcal{C}_i$. The final refined guideline $g_i$ of disease $e_i$ can be obtained by:

$$g_i = \text{LLM}([\text{Prompt}, c_{i,1}, \cdots, c_{i,k}]), \tag{2}$$

where $c_{i,j} \in \mathcal{C}_i$ is the $j$-th document. This process yields standardized, well-structured diagnostic guidelines that preserve critical clinical information while eliminating noise and redundancy. In practice, we choose Qwen2.5-72B [64] as the LLM. Examples of the guideline and prompt templates are provided in Appendix B.2.

### 3.2.2 Guideline-enhanced Feature Constraint

For multimodal downstream tasks, our framework utilizes two modality-specific encoders to separately learn visual and textual representations. The refined guideline $g$ obtained in Section 3.2.1 is employed here as the feature constraint of both textual and visual representation.

Given a sample $(x_i, t_i, y_i)$, we use the vision encoder denoted as $\Phi_{\text{img}}(\cdot)$ to extract the visual embeddings $\mathbf{V}_i$ from $x_i$. The text encoder $\Phi_{\text{text}}(\cdot)$ is employed to obtain the textual embeddings $\mathbf{T}_i$. Meanwhile, the refined guideline $g$ is also encoded by the text encoder for subsequent feature alignment. The encoding process is summarized as follows:

$$\mathbf{V}_i = \Phi_{\text{img}}(x_i) \in \mathbb{R}^{h \times w \times d}, \quad \mathbf{T}_i = \Phi_{\text{text}}(t_i) \in \mathbb{R}^{l \times d}, \quad \mathbf{G} = \Phi_{\text{text}}(g) \in \mathbb{R}^{m \times l \times d}, \tag{3}$$

where $h, w$ are the height, width of the image, $l$ is the max token length of the text encoder, $m$ is the number of disease types, and $d$ is the embedding dimension. These embeddings with spatial information are then used as the input of the dual decoder in Section 3.2.3 for multimodal fusion and final diagnosis. Here, we perform pooling on the extracted embeddings and use the pooled features for subsequent feature alignment. Specifically, we apply adaptive pooling operations to get the visual feature $\mathbf{V}_i^{'} \in \mathbb{R}^d$, and directly use the embedding of the [CLS] token as the textual feature $\mathbf{T}_i^{'} \in \mathbb{R}^d$. The corresponding pooled disease-guideline prototypes are $\mathbf{G}^{'} = \{\mathbf{G}_i^{'} \in \mathbb{R}^d \mid i = 1, 2, \ldots, m\}$.

To align the extracted features with diagnostic criteria, we propose a guideline-enhanced multi-modal feature constraint strategy. Specifically, disease-guideline prototypes are utilized as an anchor to pull both image and text features closer to them. To achieve this, we introduce a Guideline-Enhanced Contrastive Loss (GECL) for feature extraction under the guideline constraint. For sample $i$ with the disease label $y_i$, the guideline features $\mathbf{G}^{'}$ are split into $\mathbf{P}_i$ and $\mathbf{N}_i$, where $\mathbf{P}_i = \{\mathbf{G}_j^{'} \in \mathbf{G}^{'} \mid y_{ij} = 1\}$ is the set of guideline features corresponding to positive disease labels, $\mathbf{N}_i$ is the set of guideline features with negative disease labels. To avoid using excessive negative samples, we sample a subset $\mathbf{Q}_i$ from $\mathbf{N}_i$ that satisfies $|\mathbf{Q}_i| = min(r|\mathbf{P}_i|, |\mathbf{N}_i|)$, where $r$ is the negative sampling ratio. The final guideline feature set is $\mathbf{S}_i = \mathbf{P}_i \cup \mathbf{Q}_i$. Then, we can formalize GECL as a cross-entropy-based supervised contrastive learning objective:

$$\mathcal{L}_{\text{SupCon}}(\mathcal{I}_i, \mathcal{S}_i) = -\frac{1}{|\mathcal{S}_i|} \sum_{\mathcal{S}_{ij} \in \mathcal{S}_i} \left( \frac{y_{ij}}{|\mathbf{P}_i|} \phi(\mathcal{I}_i, \mathcal{S}_{ij}) - \log(1 + e^{\phi(\mathcal{I}_i, \mathcal{S}_{ij})}) \right), \tag{4}$$

$$\mathcal{L}_{\text{GECL}} = \frac{1}{N} \sum_{i=1}^{N} \left( \mathcal{L}_{\text{SupCon}}(\mathbf{T}_i^{'}, \mathbf{S}_i) + \alpha \mathcal{L}_{\text{SupCon}}(\mathbf{V}_i^{'}, \mathbf{S}_i) \right) \cdot \mathbb{I}[|\mathbf{P}_i| > 0], \tag{5}$$

where $\mathbb{I}[\cdot]$ is the indicator function. $\phi(\mathcal{I}_i, \mathcal{S}_{ij}) = \mathcal{I}_i^{\top} \mathcal{S}_{ij} / \tau$ is the similarity score between the modality-specific feature and the guideline feature, $\tau$ is the temperature hyperparameter, and $\alpha$ is the trade-off hyperparameter. Note that the similarity score $\phi$ can be converted into a probability via the Sigmoid function. As shown in Eq. (5), $\mathcal{L}_{\text{GECL}}$ aligns image features $\mathbf{V}_i^{'}$ and text features $\mathbf{T}_i^{'}$ with disease guideline prototypes, which are the diagnostic criteria of each disease defined by the embedding of its guideline. Dynamically aligning sample features with their positive prototypes prevents representation collapse while enhancing model robustness in multi-label scenarios. Furthermore, this approach induces the model to selectively focus on clinically relevant features that match the guidelines, improving the model performance and interpretability simultaneously. For detailed derivation from the standard cross-entropy form to Eq. (4), please refer to Appendix B.3.

### 3.2.3 Dual Diagnostic Network

Under the guideline constraint, we obtained enhanced visual and textual features. To achieve the final disease diagnosis, we develop a transformer-based cross-modal information fusion module, which has a dual decoder architecture. In the first decoder, the guideline $g$ is employed as the query, and the concatenated modality embeddings $\mathbf{V}_i \oplus \mathbf{T}_i$ are used as the key and value. After forward through the fusion structure $\Phi_D^g$, we obtain the logits corresponding to each disease:

$$\hat{y}_i^{\text{guide}} = \Phi_D^g(\Phi_{\text{text}}(g), \mathbf{V}_i \oplus \mathbf{T}_i, \mathbf{V}_i \oplus \mathbf{T}_i). \tag{6}$$

To further enhance the performance, we symmetrically utilize the second similar structure $\Phi_D^l$ where the query is replaced with the disease names, while keeping the key and value unchanged. This symmetric operation gets $\hat{y}_i^{\text{label}} = \Phi_D^l(\Phi_{\text{text}}(E), \mathbf{V}_i \oplus \mathbf{T}_i, \mathbf{V}_i \oplus \mathbf{T}_i) \in \mathbb{R}^m$, ensuring comprehensive feature integration. Finally, we compute the binary-cross-entropy loss on both logits with the ground truth. Thus, the total training loss of RAD is:

$$\mathcal{L}_{\text{total}} = \underbrace{\frac{1}{N}\sum_{i=1}^{N}\mathcal{L}_{\text{BCE}}(\hat{y}_i^{\text{guide}}, y_i)}_{\text{guideline branch}} + \underbrace{\frac{1}{N}\sum_{i=1}^{N}\mathcal{L}_{\text{BCE}}(\hat{y}_i^{\text{label}}, y_i)}_{\text{label branch}} + \beta\mathcal{L}_{\text{GECL}} \tag{7}$$

where $\beta$ represents the trade-off hyperparameter between $\mathcal{L}_{\text{BCE}}$ and $\mathcal{L}_{\text{GECL}}$.

## 3.3 Interpretability Evaluation System

To validate the evidence-based diagnosis of RAD, we introduce a dual-axis interpretability evaluation system that quantitatively measures the model's adherence to injected guidelines through both textual and visual metrics. Formal definitions of the metrics for each input modality are presented below.

### 3.3.1 Textual Recall of Indicators

The Guideline Recall is designed to quantify the model's explicit compliance with disease-specific diagnostic standards. The refined guideline of each disease contains a set of key laboratory indicators that are considered valuable for diagnosing this disease. The extent to which a model attends to these indicators can reflect its adherence to the guideline. Formally, when the input text contains indicators mentioned in the guideline, we assess the model's attention to these indicators by aggregating the attention weights of the corresponding tokens (derived from the cross-attention maps in the transformer

---

**Algorithm 1** Guideline Recall

1: **Input:** Guideline $G$, text token sequence $T$, attention weights $A$, threshold $\theta$
2: $\mathcal{U} \leftarrow$ Extract indicators from $G$
3: $attended = 0$, $total = 0$
4: **for** each $u \in \mathcal{U}$ **do**
5: $\quad Matched \leftarrow$ Tokens in $T$ matching $u$
6: $\quad$ **if** $Matched \neq \emptyset$ **then**
7: $\quad\quad total = total + 1$
8: $\quad\quad$ **if** mean$(A_{Matched}) > \theta$ **then**
9: $\quad\quad\quad attended = attended + 1$
10: **return** $attended/total$ if $total > 0$ else $0$

---

decoders). When the aggregated attention weights exceed a predefined threshold $\theta$, this provides quantitative evidence that the model exhibits statistically significant attention to the corresponding indicator. The detailed computation process is outlined in Algorithm 1.

### 3.3.2 Visual Attention Grounding Ability

For visual explainability, an attention-derived localization metric is employed to measure the alignment between model-attended regions and pathological abnormalities. Given expert-annotated bounding boxes for lesions, we compute the overlap between top-activated regions in the attention map and these ground truths. Specifically, we use the Intersection over Union IoU $= \frac{|A \cap B|}{|A \cup B|}$ as the metric, where $A$ is the model localization derived from the attention map and $B$ is the ground truth.

These two metrics formally establish a dual-modality interpretability evaluation system. Through the systematic analysis of how the injected knowledge explicitly intervenes in the model's decision-making, this system provides a quantitative evaluation for explainable multimodal medical AI.

Table 1: Performance across four datasets of different anatomies. The values of "Acc" and "Acc-S" on FairVLMed are the same since the dataset has only one disease. Subscript with arrows represents the absolute difference between RAD and the second-best method. $\Delta$ is the variance of RAD.

| Dataset | Method | F1 | Precision | Recall | AUC | mAP | Acc | Acc-S | Avg |
|---|---|---|---|---|---|---|---|---|---|
| MIMIC-ICD53 (Chest) | MedFuse | 34.46 | 31.36 | 45.04 | 90.85 | 31.77 | 95.34 | 41.44 | 52.89 |
| | BiomedCLIP | 32.99 | 29.56 | 45.04 | 88.71 | 29.91 | 94.72 | 39.83 | 51.54 |
| | KAD | 36.32 | 33.80 | 48.33 | 91.95 | 33.54 | 95.12 | 40.27 | 54.19 |
| | DrFuse | 34.10 | 33.70 | 45.34 | 89.50 | 31.19 | 94.68 | 38.25 | 52.39 |
| | HEALNet | 35.42 | 32.76 | 47.95 | 88.80 | 31.97 | 94.90 | 40.10 | 53.13 |
| | RAD | $\mathbf{39.71}_{3.39\uparrow}$ | $\mathbf{39.07}_{5.27\uparrow}$ | $\mathbf{54.74}_{6.41\uparrow}$ | $\mathbf{93.00}_{1.05\uparrow}$ | $\mathbf{36.74}_{3.20\uparrow}$ | $\mathbf{95.40}_{0.06\uparrow}$ | $\mathbf{42.33}_{0.89\uparrow}$ | $\mathbf{57.28}_{3.09\uparrow}$ |
| | $\Delta$ | $\pm 0.0101$ | $\pm 0.0099$ | $\pm 0.0016$ | $\pm 0.0103$ | $\pm 0.0116$ | $\pm 0.0050$ | $\pm 0.0228$ | $\pm 0.0089$ |
| FairVLMed (Eye) | MedFuse | 81.33 | 76.13 | 87.29 | 87.99 | 88.76 | 79.50 | 79.50 | 83.50 |
| | BiomedCLIP | 81.27 | 72.87 | 91.88 | 87.69 | 87.62 | 78.35 | 78.35 | 83.28 |
| | KAD | 81.18 | 73.92 | 90.03 | 88.62 | 88.88 | 78.65 | 78.65 | 83.55 |
| | DrFuse | 81.69 | 73.72 | 91.59 | 89.33 | 90.38 | 79.00 | 79.00 | 84.29 |
| | HEALNet | 81.80 | 75.22 | 89.64 | 89.60 | 90.45 | 79.60 | 79.60 | 84.39 |
| | RAD | $\mathbf{84.30}_{2.50\uparrow}$ | $\mathbf{77.52}_{1.39\uparrow}$ | $\mathbf{92.38}_{0.50\uparrow}$ | $\mathbf{91.32}_{1.72\uparrow}$ | $\mathbf{91.88}_{1.43\uparrow}$ | $\mathbf{82.40}_{2.80\uparrow}$ | $\mathbf{82.40}_{2.80\uparrow}$ | $\mathbf{86.63}_{2.24\uparrow}$ |
| | $\Delta$ | $\pm 0.0028$ | $\pm 0.0070$ | $\pm 0.0005$ | $\pm 0.0126$ | $\pm 0.0144$ | $\pm 0.0080$ | $\pm 0.0080$ | $\pm 0.0060$ |
| SkinCAP (Skin) | MedFuse | 79.25 | 85.96 | 77.99 | 96.50 | 73.61 | 99.34 | 74.36 | 83.86 |
| | BiomedCLIP | 81.49 | 87.13 | 81.41 | 97.22 | 79.22 | 99.11 | 74.36 | 85.71 |
| | KAD | 82.06 | 86.79 | 81.27 | 97.80 | 80.40 | 99.25 | 75.46 | 86.15 |
| | DrFuse | 81.18 | 85.70 | 79.64 | 94.92 | 76.42 | 99.29 | 77.66 | 84.97 |
| | HEALNet | 82.20 | 88.69 | 81.18 | 92.68 | 77.97 | 99.37 | 78.39 | 85.79 |
| | RAD | $\mathbf{85.48}_{3.28\uparrow}$ | $\mathbf{89.48}_{0.79\uparrow}$ | $\mathbf{83.23}_{1.82\uparrow}$ | $\mathbf{97.97}_{0.17\uparrow}$ | $\mathbf{83.55}_{3.15\uparrow}$ | $\mathbf{99.48}_{0.14\uparrow}$ | $\mathbf{81.32}_{2.93\uparrow}$ | $\mathbf{88.64}_{2.49\uparrow}$ |
| | $\Delta$ | $\pm 0.0678$ | $\pm 0.0750$ | $\pm 0.0136$ | $\pm 0.0356$ | $\pm 0.0639$ | $\pm 0.0159$ | $\pm 0.0474$ | $\pm 0.0407$ |
| NACC (Brain) | MedFuse | 31.53 | 25.59 | 68.36 | 85.50 | 24.49 | 87.44 | 58.45 | 54.48 |
| | BiomedCLIP | 34.36 | 29.02 | 66.95 | 84.00 | 26.03 | 88.80 | 58.21 | 55.34 |
| | KAD | 35.09 | 29.68 | 64.49 | 85.88 | 27.73 | 89.69 | 57.86 | 55.77 |
| | DrFuse | 34.11 | 27.86 | $\mathbf{68.96}$ | 82.88 | 27.88 | 87.99 | 51.31 | 54.43 |
| | HEALNet | 35.91 | 28.92 | 67.33 | 85.04 | 26.13 | 89.55 | 56.79 | 55.67 |
| | RAD | $\mathbf{37.65}_{1.74\uparrow}$ | $\mathbf{36.24}_{7.32\uparrow}$ | $65.78_{1.55\downarrow}$ | $\mathbf{87.11}_{2.07\uparrow}$ | $\mathbf{30.03}_{3.90\uparrow}$ | $\mathbf{90.36}_{0.81\uparrow}$ | $\mathbf{59.64}_{2.85\uparrow}$ | $\mathbf{58.12}_{2.45\uparrow}$ |
| | $\Delta$ | $\pm 0.0015$ | $\pm 0.0049$ | $\pm 0.0003$ | $\pm 0.0019$ | $\pm 0.0023$ | $\pm 0.0010$ | $\pm 0.0078$ | $\pm 0.0020$ |

# 4 Experiments

## 4.1 Experimental Setup

**Datasets.** We evaluate RAD on four multimodal medical datasets with different anatomies, including MIMIC-ICD53, Harvard-FairVLMed [41], SkinCAP [76], and NACC [5].

Table 2: Detailed information of the datasets.

| Dataset | Anatomy | Modality | Label | Sample |
|---|---|---|---|---|
| MIMIC-ICD53 | Chest | X-ray Image & Report & EHR (Lab Results) | 53 | 51830 |
| Harvard-FairVLMed | Eye | Fundus Image & Report & Demographics | 1 | 10000 |
| SkinCAP | Skin | Dermatology Image & Report | 50 | 2526 |
| NACC | Brain | 3D MRI Image & EHR (Lab Results) | 11 | 4199 |

MIMIC-ICD53 is constructed through the alignment and integration of MIMIC-CXR [28] and MIMIC-IV [29], comprising chest X-ray images, corresponding reports, and EHRs, annotated with 53 diseases under the ICD [43] standard. For laboratory indicators in the EHR, we quantified the numerical results on a scale of 1 to 10 based on the upper and lower limits of their normal range. We will release the dataset on PhysioNet [42]. Details of dataset construction are provided in Appendix C.1.1. Harvard-FairVLMed, SkinCAP, and NACC are multimodal datasets focusing on eyes, skin, and brain, respectively. All patient data has been de-identified. More detailed statistics of datasets are presented in Table 2.

**Baselines.** We select representative baseline methods in the medical field, including large-scale pre-training model *BiomedCLIP* [70], knowledge-enhanced pre-training method *KAD* [71], and state-of-the-art multimodal fusion methods *MedFuse* [21], *DrFuse* [66], and *HEALNet* [23].

**Evaluation Metrics.** For the evaluation of model performance, we adopt widely used multi-label classification metrics including F1, Precision, Recall, AUC, mAP, and ACC. All metrics are the average of multiple labels. Since standard accuracy (ACC) aggregates predictions across all labels and thus may not adequately reflect the correctness for individual patients, we include an additional metric named sample-wise ACC (ACC-S). This metric considers a prediction correct only if all labels of a patient are accurately classified, making it more aligned with clinical scenarios.

Table 3: Quantitative evaluation of Visual Explainability. We calculate the metrics for each disease category and report both disease-averaged (Avg-D) and patient-averaged (Avg-P) values.

| Method | Visual Grounding (mIoU) | | | | | | | | |
|---|---|---|---|---|---|---|---|---|---|
| | Consolidation | Atelectasis | Effusion | Emphysema | Fibrosis | Fracture | Mass | Avg-D | Avg-P |
| w/o RAD | 17.68 | 19.23 | 18.89 | 14.95 | 17.22 | 13.13 | 10.81 | 15.98 | 17.78 |
| RAD | 24.30 | 20.74 | 20.13 | 21.15 | 19.42 | 17.14 | 15.15 | 19.72 | 22.04 |

Table 4: Quantitative evaluation of Textual Explainability. We present the guideline recall on representative laboratory indicators and the total average recall. The indicator names are abbreviated.

| Method | Guideline Recall | | | | | | |
|---|---|---|---|---|---|---|---|
| | PC | Bilirubin | ALT | IBC | WBC | AST | Total |
| w/o RAD | 23.82 | 31.34 | 6.81 | 37.38 | 11.96 | 4.41 | 24.76 |
| RAD | 64.55 | 51.71 | 57.96 | 71.82 | 29.09 | 40.65 | 65.62 |

**Implementation Details.** In practice, Top-$k$ in Eq.(1) is set to 10. All guidelines obtained by Eq.(2) and the indicators used in Algorithm 1 are manually verified to avoid potential factual errors. The default backbone of the text image encoder is ClinicalBERT [55] and ResNet-50 [22], respectively. The hyperparameters $\alpha$ and $\beta$, which serve as the balancing ratio between different losses, are set to be $1e-2$ and $1e-1$, respectively. All experiments are conducted on a single NVIDIA A100 GPU.

## 4.2 Diagnosis Performance

As demonstrated in Table 1, our method consistently achieves superior performance across four benchmarks of diverse anatomies. Specifically, RAD outperforms the second-best method with average improvements of 3.09%, 2.24%, 2.49%, and 2.45% on MIMIC-ICD53, FairVLMed, Skin-CAP, and NACC datasets, respectively. The most substantial gains occur in MIMIC-ICD53, where RAD improves both precision and recall over 5%, suggesting strong robustness in handling complex, real-world clinical label distributions. This improvement is particularly noteworthy given the dataset's challenging nature, containing both fine-grained ICD labels and noisy clinical documentation. Notably, the sample-wise accuracy (ACC-S) of all methods exhibits a significant degradation compared to macro-average accuracy (ACC), especially in datasets with extensive label spaces. This discrepancy highlights fundamental limitations in current models' capacity to handle multi-label problems, exposing challenges for real-world clinical deployment. Intriguingly, KAD, which injects medical knowledge during the pretraining phase, achieves strong performance on MIMIC-ICD53 but falls short on others. This is likely because its pretraining data concentrated on chest X-rays, limiting its ability to generalize to other anatomical regions. In contrast, our approach directly injects knowledge on downstream tasks, offering greater adaptability across distinct regions and modalities. These consistent improvements across diverse anatomies, data scales, and label complexities validate the versatility and scalability of RAD. Full baseline results with variance are in Appendix C.3.

## 4.3 Interpretability Evaluation

### 4.3.1 Interpretability from Textual Perspective

To quantitatively assess the impact of knowledge injection from the textual perspective, we calculate the guideline recall defined in Section 3.3.1. As presented in Table 4, incorporating knowledge via RAD prominently increases the recall value from 24.76% to 65.62%. This indicates that RAD indeed injects guideline-derived knowledge into the model, thereby enhancing its focus on key information mentioned in the guideline. Notably, the conventional model exhibits extremely low recall (<10%) on Alanine Aminotransferase (ALT) and Aspartate Aminotransferase (AST). This may stem from their inability to understand highly specialized, rare medical terms. In contrast, RAD explicitly highlights the importance of these indicators in the guideline, leading to significant recall improvement. This finding underscores the necessity of flexible knowledge adaptation for downstream tasks rather than static pretraining paradigms. Overall, the enhanced guideline recall demonstrates that RAD enables the model to make reliable evidence-based diagnoses according to guidelines. This improvement also aligns with the qualitative attention patterns observed in Figure 1. To further substantiate the interpretability of RAD, we provide more and clearer visualization cases in Appendix C.5.

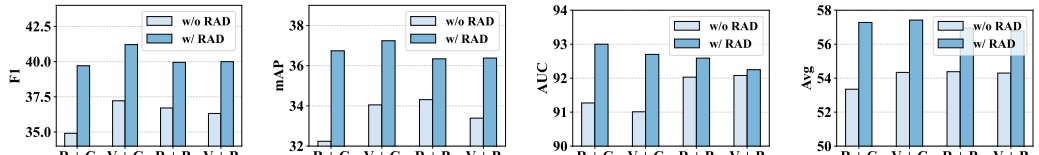

Figure 4: Performance under different combinations of modality encoder backbones on the MIMIC-ICD53 dataset. (R = ResNet, V = ViT, C = ClinicalBERT, B = BioClinicalBERT)

Table 5: Ablation on each component of our method. "×" in the "Decoder" column means replacing our dual diagnostic decoder with a conventional MLP. The best results are in **boldface**.

| $\mathcal{L}_{\text{GECL}}^{\text{vision}}$ | $\mathcal{L}_{\text{GECL}}^{\text{text}}$ | Decoder | F1 | Precision | Recall | AUC | mAP | Acc | Acc-S | Avg |
|---|---|---|---|---|---|---|---|---|---|---|
| × | × | × | 34.91 | 31.01 | 50.91 | 91.27 | 32.24 | 94.50 | 38.63 | 53.35 |
| ✓ | × | × | 37.43 | 33.98 | 51.44 | 92.53 | 34.80 | 95.26 | 38.10 | 54.79 |
| × | ✓ | × | 37.75 | 36.32 | 51.52 | 92.91 | 35.03 | 95.43 | 39.65 | 55.52 |
| ✓ | ✓ | × | 39.34 | 37.74 | 51.87 | 92.94 | 36.36 | **95.59** | 39.95 | 56.26 |
| × | × | ✓ | 39.22 | 36.88 | 51.41 | 92.25 | 36.44 | 95.39 | 39.80 | 55.91 |
| ✓ | ✓ | ✓ | **39.71** | **39.07** | **54.74** | **93.00** | **36.74** | 95.40 | **42.33** | **57.28** |

### 4.3.2 Interpretability from Visual Perspective

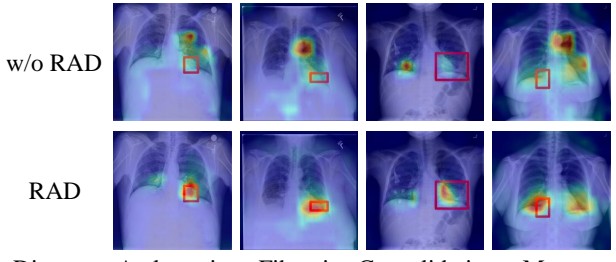

Figure 3: Visualization of grounding results on four diseases.

Symmetrically, we conduct zero-shot grounding experiments on the ChestX-Det dataset [38]. The results shown in Table 3 demonstrate a significant improvement in mIoU scores for lesion detection after the injection of refined guidelines. Besides, Figure 3 illustrates multiple cases of lesion grounding. For clearer visualization, we overlay spectrum heatmaps on the original CXR images, together with the ground truth bounding box highlighted in red. A comparison between the lesions identified by the model and the bounding boxes marked by clinical experts reveals a notable improvement in alignment when our guidelines are applied. This indicates that the model's focus is more accurately directed toward clinically significant lesions, emphasizing RAD's enhanced diagnosis capabilities and interpretability under the guidance of external knowledge.

### 4.4 Ablation Study

In this subsection, we conduct ablation studies on each component of RAD and validate its generalizability across different model architectures. All experiments are conducted on MIMIC-ICD53.

**Ablation on Each Component.**  As shown in Table 5, we evaluate the efficacy of each newly proposed component in RAD. It is evident that removing either the $\mathcal{L}_{\text{GECL}}$ or the dual decoder negatively impacts model performance, highlighting the importance of the guideline in both representation learning and multimodal fusion. Notably, the removal of the Dual Decoder results in the most substantial performance degradation, underscoring the necessity of leveraging guidelines to intervene in the final decision-making process. We further compared the performance of the textual and visual branches of $\mathcal{L}_{\text{GECL}}$ when used individually. The results show that the textual branch yields more significant improvements. This can be attributed to the fact that both the input text and the guideline belong to the same modality, allowing for more effective alignment.

**Ablation on Different Backbones.**  To demonstrate the robustness and flexibility of our method, we verify RAD on different encoder backbone combinations. Specifically, we iteratively replaced the default visual encoder and text encoder with two other popular architectures, ViT [15] and

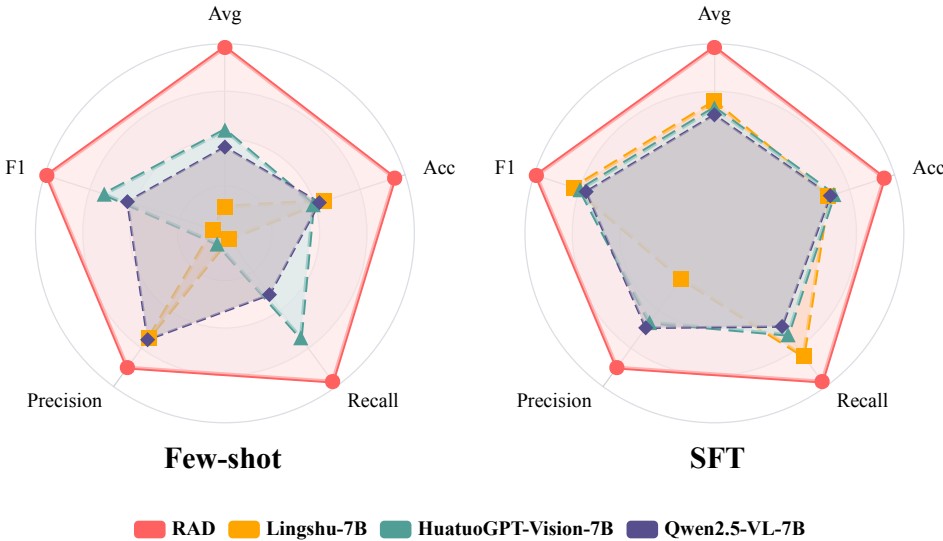

Figure 5: Performance comparison with MLLMs. We convert the single-label dataset FairVLMed into the VQA format and evaluate MLLMs under both few-shot and supervised fine-tuning settings.

BioClinicalBERT [2]. As illustrated in Figure 4, RAD consistently offers substantial performance gain across all four combinations of backbones. This not only highlights the insensitivity of our approach to different backbone architectures but also underscores its robustness and generalizability. Specifically, ResNet and ViT exhibit comparable performance gains, while ClinicalBERT shows more pronounced improvement than BioClinicalBERT. Overall, RAD exhibits the robust ability to generalize to diverse data and model structures, ensuring reliable performance in various scenarios. Detailed results, more ablation studies, and hyperparameter analysis are presented in Appendix C.6.

### 4.5 Discussion

**Comparison with Multimodal Large Language Models** Multimodal large language models exhibit remarkable capabilities in visual content understanding and generalization. To further validate the effectiveness of RAD, we compare with state-of-the-art MLLMs, including Qwen2.5-VL-7B [3], HuatuoGPT-Vision-7B [10], and Lingshu-7B [62]. As presented in Figure 5, our discriminative framework achieves superior performance with significantly lower computational cost. These results demonstrate that complex diagnostic tasks are better suited for specialized discriminative models than generative MLLMs. The significant performance gap, observed on the simplest single-label dataset, underscores the practical advantages of our approach in clinical applications where both accuracy and efficiency are critical.

## 5 Conclusion

This paper proposes RAD, which enhances the capabilities of multimodal diagnosis models by leveraging external medical knowledge. RAD operates via a tri-fold methodology, consisting of offline retrieval and refinement of disease-centered external guidelines, multimodal feature alignment under the guideline constraint, and the dual diagnostic network. Extensive experiments on four datasets of different anatomies demonstrate the effectiveness of RAD. Furthermore, RAD exhibits dual-axis interpretability by simultaneously achieving precise lesion localization in imaging data and prioritizing guideline-concordant indicators in textual analysis. This evidence-based explainability enhances clinical trustworthiness, offering the potential to inspire future research in medical AI.

## Acknowledgement

This work is supported by the National Key R&D Program of China (No. 2022ZD0160703), National Natural Science Foundation of China (No. 62306178) and STCSM (No. 22DZ2229005), 111 plan (No. BP0719010), and Shanghai Artificial Intelligence Laboratory.

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

# Appendix

# A  Further Discussion

## A.1  Broader impact

The method proposed in this paper can effectively enhance the diagnostic capability of multimodal medical models. With the integration of the guidelines, RAD is optimized through intervention in accordance with the guidelines. This not only improves diagnostic accuracy but also strengthens the model's interpretability, making its decision-making process more transparent and deployable in real-world clinical scenarios. Specifically, the systematic integration of multimodal data (imaging, text, and structured records) enables RAD to capture disease manifestations from multiple perspectives, potentially advancing personalized medicine through comprehensive patient profiling. However, the use of multimodal clinical data, including sensitive patient records and imaging features, necessitates stringent compliance with relevant regulations to prevent misuse or unintended leakage of private health information.

## A.2  Limitations

A limitation of the current implementation is the static retrieval knowledge corpus. While medical guidelines undergo periodic updates (e.g., every 3-5 years) to incorporate new evidence and diseases, RAD relies on a fixed knowledge base that may require manual updates to reflect revised diagnostic standards. This temporal mismatch could be addressed by regular updates of guidelines and further fine-tuning of the models, which would enhance long-term clinical relevance without compromising current performance.

# B  Method Details

## B.1  Knowledge-corpus Construction

For retrieving disease-related diagnostic knowledge, we collect medical knowledge from four distinct sources: "Wiki", "Research", "Guideline", and "Book". **Wikipedia** provides comprehensive and general descriptions of target diseases, such as standard disease nomenclature, formal medical definitions, and clinically relevant subcategories. The processed data are obtained from Huggingface[2]. **Research** incorporates the latest research articles from PubMed (a premier database of biomedical literature). These articles provide cutting-edge findings in disease mechanisms, diagnostic criteria, and therapeutic interventions. We utilize the 2024 PubMed baseline[3], which is a complete snapshot of PubMed data. We filter the valid data through their paper titles and corresponding abstracts. **Guideline** includes 45K clinical practice guidelines from 13 sources. The guidelines provide rigorously vetted diagnostic criteria and treatment protocols for medical practitioners, serving as a critical component for reliable decision support. We employ the Clinical Guidelines dataset [12] and use the provided scripts to crawl non-redistributable portions of the data. **Book.** consists of diverse medical textbooks. These books cover well-organized basic medical knowledge in surgery, medical imaging, and drugs, etc. We follow MedOmniKB [13] to collect 18K PDF documents from online medical libraries and academic publishers. Then, deduplicate and filter these books to obtain the final retrieval database.

## B.2  Details for LLM Refinement

An example of the final guideline is presented in Figure 6. For more guidelines, please refer to our GitHub repository.

The detailed prompt template for LLM refinement is shown in Figure 7. In the prompt, {disease_icd_name} is the disease name $e_i$, {topk} is the number of preserved documents, and {retrieve_passages_str} is the content of the document.

---

[2] https://huggingface.co/datasets/wikimedia/wikipedia
[3] https://ftp.ncbi.nlm.nih.gov/pubmed/baseline

Summary of Key Diagnostic Features for Bronchitis

Disease Description: Bronchitis is an inflammation of the bronchi, the air passages in the lungs. It can be classified into two main types: acute and chronic. Acute bronchitis is typically a self-limiting condition characterized by a cough that may produce sputum and is often caused by viral infections. Chronic bronchitis, on the other hand, is a long-term condition defined by a productive cough lasting for at least three months in two consecutive years, often associated with chronic obstructive pulmonary disease (COPD). The primary risk factor for chronic bronchitis is tobacco smoking, with other factors including air pollution and occupational exposures.

Important Lab Tests and Values: - Acute Bronchitis: White Blood Cell Count (WBC): Usually normal or slightly elevated. C-reactive Protein (CRP): May be slightly elevated but not typically high. Sputum Culture: Not routinely necessary, but can be useful if bacterial infection is suspected. Chronic Bronchitis: Pulmonary Function Tests (PFTs): Reduced FEV1/FVC ratio, indicating airflow obstruction. Sputum Analysis: Increased mucus production, often with neutrophil infiltration. Blood Gas Analysis:** May show hypoxemia and hypercapnia in advanced cases.

Key Radiological or Clinical Findings: Acute Bronchitis: Chest X-ray: Usually normal, but may show hyperinflation or peribronchial thickening. Physical Examination: Wheezing, crackles, and rhonchi on auscultation. Chronic Bronchitis: Chest X-ray: May show hyperinflation, increased bronchovascular markings, and signs of emphysema. CT Scan: Can reveal bronchial wall thickening and mucus plugging. Physical Examination: Barrel chest, cyanosis, and signs of cor pulmonale in advanced cases.

Diagnostic Symptoms or Relevant Clinical Features: Acute Bronchitis: Cough: Initially dry, then becomes productive with clear or yellowish sputum. Fever: Usually mild or absent; high fever suggests pneumonia. Fatigue and Body Aches: Common but generally mild. Wheezing and Shortness of Breath: May be present, especially in patients with underlying asthma. Chronic Bronchitis: Cough: Persistent, productive cough with sputum, often for at least three months in two consecutive years. Dyspnea: Shortness of breath, especially on exertion. Wheezing: Common, especially in the morning. Chest Pain: May occur due to prolonged coughing. Fatigue and Malaise: Persistent, often due to chronic hypoxemia.

Figure 6: Examples of the guidelines.

## B.3 Derivation of the Guideline-Enhanced Contrastive Loss

Here, we derive the form of the most basic cross-entropy loss to the form in Eq. (4). We demonstrate the equivalence between the sigmoid-based cross-entropy formulation and the logit-style implementation of our supervised contrastive loss.

The equation in cross-entropy form with explicit sigmoid terms is defined as:

$$\mathcal{L}_{\text{SupCon}}(\mathcal{I}_i, \mathcal{S}_i) = \frac{1}{|\mathcal{S}_i|} \sum_{j \in \mathcal{S}_i} \left[ \frac{y_{ij}}{|\mathbf{P}_i|} \log \sigma(\phi_{ij}) + \left(1 - \frac{y_{ij}}{|\mathbf{P}_i|}\right) \log(1 - \sigma(\phi_{ij})) \right], \quad (8)$$

where $\phi_{ij}$ is short for $\phi(\mathcal{I}_i, \mathcal{S}_{ij}) = \mathcal{I}_i^\top \mathcal{S}_{ij}/\tau$, the similarity score between the modality-specific feature and the guideline feature. $\sigma(\cdot)$ is the Sigmoid function, the logit of the similarity score is $\sigma(\phi_{ij}) = \frac{1}{1+e^{-\phi_{ij}}}$. The difference between Eq. (8) and the standard cross-entropy loss is that we use the similarity score as logits, and we add the normalization coefficient $\frac{1}{|\mathbf{P}_i|}$ to balance the gradient contribution of each positive label in multi label scenarios. The following is a step-by-step derivation. First, substitute the sigmoid function into Eq. (8):

> **Prompt for LLM Refinement:**
>
> Your task is to help filter and summarize the relevant information from multiple sets of retrieved external knowledge (from 4 different sources) to support my multi-modal disease classification model in diagnosing the specific disease associated with the provided ICD disease description. For the given disease, identify the critical symptoms, lab indicators, and radiological features that are most strongly associated with the disease diagnosis. Discard any information that is unrelated or irrelevant to disease classification. You do not need to focus on treatment options, but instead, concentrate on factors that would help in diagnosing the disease. You will be provided with a set of documents (containing topk retrieved documents, each approximately 2000 characters), containing a range of medical information. Your job is to:
>
> Review the retrieved documents and determine which information is directly relevant to diagnosing the disease, based on its ICD code and description. Eliminate any information that is unrelated to diagnosis or classification, such as treatment options, management strategies, or irrelevant clinical details. Focus on identifying key diagnostic features, including symptoms, laboratory test results, and imaging findings that help confirm the presence of the disease. Evaluate the importance of each retrieved documents: Some documents may provide more critical or reliable information than others. Prioritize information that is most relevant and useful for diagnosing the disease, even if it means excluding less relevant details from certain documents. Summarize the most relevant content into a single cohesive summary of approximately 2000 characters (or 500 words). The summary should include the essential diagnostic criteria, including lab values, clinical features, and radiological findings.
> The summary should emphasize:
> A brief explanation or description of the disease, including any variations or related conditions under the same ICD code. Important lab tests and values (e.g., white blood cell count, C-reactive protein). Key radiological or clinical findings associated with the disease's presence (e.g., lung opacity, pleural effusion). Any diagnostic symptoms or relevant clinical features. Discard information that is not useful for diagnosing the disease.
> Please ensure that the summary is concise and directly related to diagnosis, omitting irrelevant details.
> Disease description (ICD name): {disease_icd_name}
> Retrieved passages: {retrieve_passages_str}

Figure 7: Prompt for LLM refinement.

$$\mathcal{L}_{\text{SupCon}} = \frac{1}{|\mathcal{S}_i|} \sum_j \left[ \frac{y_{ij}}{|\mathbf{P}_i|} \log \sigma(\phi_{ij}) + \left(1 - \frac{y_{ij}}{|\mathbf{P}_i|}\right) \log(1 - \sigma(\phi_{ij})) \right]$$

(Substitute sigmoid identities: $\log \sigma(\phi) = \phi - \log(1 + e^{\phi})$, $\log(1 - \sigma(\phi)) = -\log(1 + e^{\phi})$)

$$= \frac{1}{|\mathcal{S}_i|} \sum_j \left[ \frac{y_{ij}}{|\mathbf{P}_i|} \left(\phi_{ij} - \log(1 + e^{\phi_{ij}})\right) + \left(1 - \frac{y_{ij}}{|\mathbf{P}_i|}\right) \left(-\log(1 + e^{\phi_{ij}})\right) \right]$$

$$= \frac{1}{|\mathcal{S}_i|} \sum_j \left[ \frac{y_{ij}}{|\mathbf{P}_i|} \phi_{ij} - \frac{y_{ij}}{|\mathbf{P}_i|} \log(1 + e^{\phi_{ij}}) - \left(1 - \frac{y_{ij}}{|\mathbf{P}_i|}\right) \log(1 + e^{\phi_{ij}}) \right]$$

$$= \frac{1}{|\mathcal{S}_i|} \sum_j \left[ \frac{y_{ij}}{|\mathbf{P}_i|} \phi_{ij} - \log(1 + e^{\phi_{ij}}) \left(\frac{y_{ij}}{|\mathbf{P}_i|} + 1 - \frac{y_{ij}}{|\mathbf{P}_i|}\right) \right]$$

$$= \frac{1}{|\mathcal{S}_i|} \sum_j \left[ \frac{y_{ij}}{|\mathbf{P}_i|} \phi_{ij} - \log(1 + e^{\phi_{ij}}) \right]$$

The final equation is:

$$\mathcal{L}_{\text{SupCon}}(\mathcal{I}_i, \mathcal{S}_i) = -\frac{1}{|\mathcal{S}_i|} \sum_{\mathcal{S}_{ij} \in \mathcal{S}_i} \left( \frac{y_{ij}}{|\mathbf{P}_i|} \phi(\mathcal{I}_i, \mathcal{S}_{ij}) - \log(1 + e^{\phi(\mathcal{I}_i, \mathcal{S}_{ij})}) \right). \tag{9}$$

We use this form directly in the main body Section 3.2.2.

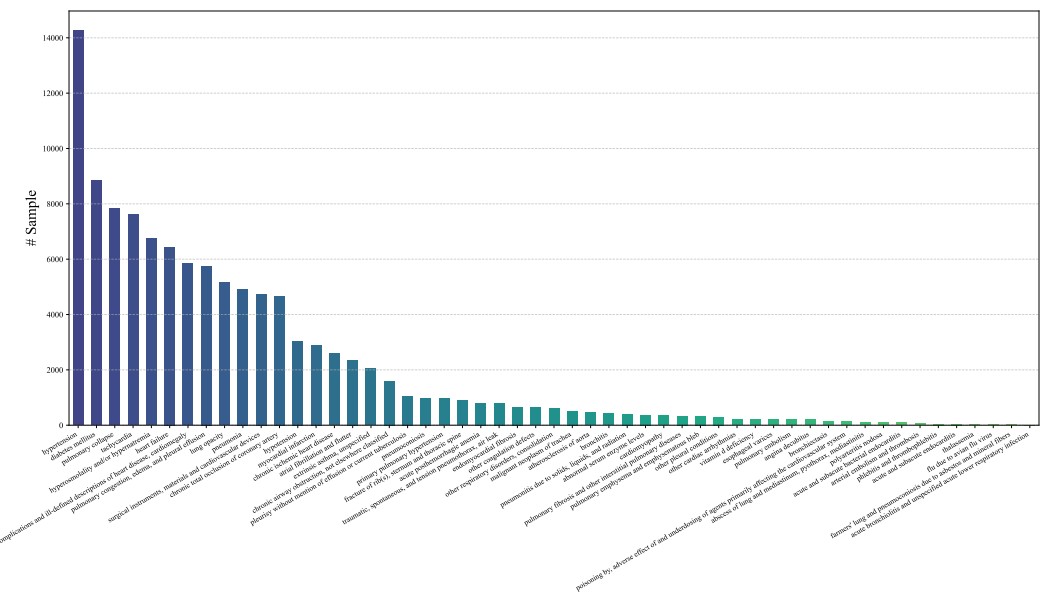

Figure 8: Label distribution of MIMIC-ICD53. The X-axis represents the formal disease names under the ICD-10 standard.

## C   Experimental Details

### C.1   Details of Datasets

#### C.1.1   Construction Process of MIMIC-ICD53

First, we merged and aligned the ED, HOSP, and ICU parts of MIMIC-IV [29]. Subsequently, we aligned the processed MIMIC-IV dataset with the MIMIC-CXR-JPG dataset [28]. We utilized patient_id and study_id to align the datasets at the patient level. Given that temporal patient dynamics were not considered, we selected the most recent radiological examination for each patient, including the associated images and reports. For temporal alignment, we extracted EHR data and ICD disease codes from MIMIC-IV within a three-day window following the radiological examination. After excluding instances with missing modalities or labels, we obtained a final sample size of 51830. For disease labeling, we standardized the granularity of diagnoses according to the ICD-10 classification, using the format $Xab$ (where $X$ is a letter and $ab$ are digits), resulting in over 2000 unique labels. To refine and further clean the dataset, we consulted LLMs and then physicians to identify and select 53 critical disease categories that were related to thoracic and cardiovascular conditions or could be identified using laboratory indicators in the EHR. For the numerical indicators in the EHR, we quantized each indicator on a 0-10 integer scale based on its corresponding normal range limits. A value of 4-7 is considered normal, 0-3 indicates too low, and 8-10 signifies too high. The label distribution of MIMIC-ICD53 is shown in Figure 8. The training set and test set are randomly divided in a ratio of 4:1. The final processed dataset, termed MIMIC-ICD53, will be made publicly available on PhysioNet after publication. (The MIMIC dataset requires that all datasets developed based on MIMIC can only be released on PhysioNet.)

#### C.1.2   Preprocess of NACC

The National Alzheimer's Coordinating Center (NACC) dataset [5] is a large, standardized resource comprising clinical and neuropathological data collected from individuals assessed at Alzheimer's Disease Research Centers (ADRCs) across US, which consists various neurodegenerative diseases, like Alzheimer's disease, Parkinson's disease, vascular dementia, and other forms of cognitive impairment. We follow [63] to organize the dataset, resulting in 11 labels including: "Normal cognition" (NC), "Mild cognitive impairment" (MCI), "Dementia" (DE), "Alzheimer's disease" (AD), "Vascular dementia, vascular brain injury and vascular dementia" (VD), "Lewy body dementia, including dementia with Lewy bodies and Parkinson's disease dementia" (LBD), "Psychiatric conditions including schizophrenia, depression, bipolar disorder, anxiety and posttraumatic stress disorder"

(PSY), "Frontotemporal lobar degeneration and its variants, including primary progressive aphasia, corticobasal degeneration and progressive supranuclear palsy, and with or without amyotrophic lateral sclerosis" (FTD), "Systemic and environmental factors including infectious diseases (HIV included), metabolic, substance abuse / alcohol, medications, systemic disease and delirium" (SEF), "Other dementia conditions, including neoplasms, Down syndrome, multiple systems atrophy, Huntington's disease and seizures" (ODE), and "Moderate/severe traumatic brain injury, repetitive head injury and chronic traumatic encephalopathy" (TBI). The label distribution of NACC is shown in Figure 9. Given that NACC contains over 800 distinct EHR variables, selecting the most relevant features for analysis was a critical step in our study. To ensure both scientific validity and clinical interpretability, we first utilized LLM for cleaning and double-checked with several physicians.

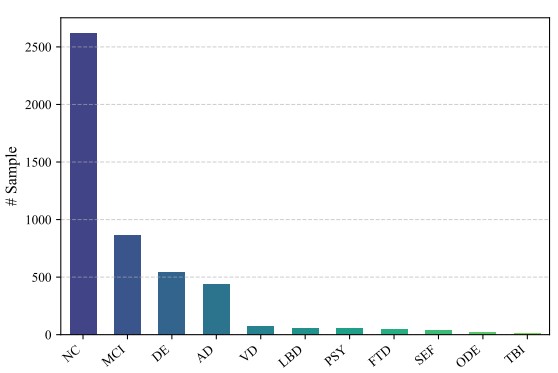

Figure 9: Label distribution of NACC.

Finally, we distilled the original set down to a final list of 36 key EHR variables. The selected variables include height, weight, body mass index (BMI), systolic blood pressure, diastolic blood pressure, cortical atrophy (Alzheimer's disease marker), small vessel disease (vascular dementia related), left motor cortex vascular lesion, right motor cortex vascular lesion, normal pressure hydrocephalus gait, parkinsonian signs (tremor/rigidity), bradykinesia (Parkinsonian symptom), neck rigidity (dystonia), gait disturbance, history of hypertension, history of diabetes, history of cardiovascular disease, history of stroke, history of Parkinson's disease, sleep apnea, REM sleep behavior disorder (RBD), history of traumatic brain injury (TBI), delusions, hallucinations, depressive symptoms, agitation or aggression, anti-dementia medication (e.g.), Parkinson's disease medication (e.g.), anticoagulant use (stroke prevention), antidepressant medication, postural instability (Parkinson's or Lewy body dementia), APOE $\epsilon 4$ allele (Alzheimer's disease risk), hypercholesterolemia (vascular risk), amyotrophic lateral sclerosis (ALS) signs, left visual cortex functional impairment, and right visual cortex functional impairment. The final processed dataset is randomly divided in a ratio of 4:1 for training and testing.

### C.1.3 Details of Harvard-FairVLMed and SkinCAP

The Harvard-FairVLMed dataset [41], sourced from the Department of Ophthalmology at Harvard Medical School, contains 10,000 multimodal samples (7,000 train, 1,000 val, 2,000 test) with paired clinical notes, diagnostic labels, and detailed demographic attributes (race, gender, ethnicity, language). The dataset is publicly available under the CC BY-NC-ND 4.0 license at Github[4]. We directly used the original dataset.

SkinCAP is a multimodal dermatology dataset containing 4,000 expert-annotated skin disease images with rich natural language descriptions [76]. The dataset combines cases from diverse dermatology image datasets, all annotated by board-certified dermatologists to ensure clinical accuracy. It is publicly available under an open license at HuggingFace[5]. To address class imbalance, we removed tail categories with too few positive samples, resulting in a filtered dataset of 2,526 samples with 50 disease labels. The final dataset was partitioned into a 4:1 train-test split.

---

[4]https://github.com/Harvard-Ophthalmology-AI-Lab/FairCLIP
[5]https://huggingface.co/datasets/joshuachou/SkinCAP

Table 6: Benchmarking performance of each modality on MIMIC-ICD53 dataset.

| Modality | Model/Method | F1 | Precision | Recall | AUC | mAP | Acc | Acc-S | Avg |
|----------|-------------|-----|-----------|--------|-----|-----|-----|-------|-----|
| Image | ResNet18 | 14.39 | 10.49 | 32.94 | 71.47 | 9.66 | 87.60 | 21.94 | 35.50 |
| | ResNet50 | 14.49 | 10.99 | 33.09 | 72.35 | 9.92 | 87.68 | 22.41 | 35.85 |
| | ViT-Base | 15.49 | 11.59 | 34.90 | 73.93 | 10.81 | 88.00 | 20.66 | 36.75 |
| | Swin-Base | 16.93 | 13.61 | 33.97 | 75.12 | 12.11 | 89.03 | 18.94 | 37.10 |
| | UniChest | 19.30 | 16.68 | 31.45 | 78.16 | 14.18 | 92.00 | 21.63 | 39.06 |
| | RadDino | 17.26 | 13.79 | 29.90 | 75.34 | 11.75 | 90.89 | 13.87 | 36.11 |
| EHR | MLP | 19.14 | 15.15 | 38.52 | 72.57 | 14.81 | 89.31 | 0.00 | 35.64 |
| | TabFPN | 9.20 | 5.85 | 53.32 | 52.70 | 5.36 | 59.24 | 0.00 | 26.52 |
| | ClinicalBERT | 14.96 | 9.99 | 50.41 | 79.80 | 9.65 | 84.16 | 18.03 | 38.14 |
| | BioClinicalBERT | 11.90 | 8.57 | 42.57 | 72.52 | 7.21 | 80.51 | 18.98 | 34.61 |
| | PubMedBERT | 10.18 | 6.31 | 78.48 | 67.16 | 5.87 | 53.35 | 19.88 | 34.46 |
| | LLaMa-3.2-1B | 22.00 | 18.94 | 35.86 | 84.47 | 16.52 | 91.92 | 15.85 | 40.79 |
| | LLaMa-3.1-8B | 22.53 | 18.55 | 38.84 | 84.98 | 16.66 | 92.28 | 17.65 | 41.64 |
| | MMedS-8B | 21.88 | 18.00 | 34.47 | 84.72 | 15.99 | 92.47 | 19.98 | 41.07 |
| Report | ClinicalBERT | 24.29 | 21.06 | 39.95 | 83.23 | 18.45 | 91.52 | 22.66 | 43.02 |
| | BioClinicalBERT | 29.12 | 25.13 | 45.18 | 84.21 | 23.03 | 91.59 | 17.29 | 45.08 |
| | PubMedBERT | 15.46 | 12.06 | 36.86 | 73.80 | 9.65 | 87.07 | 26.59 | 37.36 |
| | LLaMa-3.2-1B | 30.86 | 28.91 | 42.67 | 86.81 | 25.46 | 93.83 | 22.53 | 47.30 |
| | LLaMa-3.1-8B | 32.53 | 31.84 | 42.03 | 87.54 | 27.30 | 94.14 | 25.29 | 48.67 |
| | MMedS-8B | 32.39 | 29.52 | 43.73 | 86.78 | 27.08 | 94.17 | 22.97 | 48.09 |
| Report +EHR | ClinicalBERT | 27.13 | 23.43 | 46.32 | 89.22 | 22.20 | 92.47 | 31.32 | 47.44 |
| | BioClinicalBERT | 28.18 | 23.62 | 46.96 | 89.28 | 22.28 | 92.26 | 29.26 | 47.41 |
| | PubMedBERT | 10.10 | 6.02 | 82.24 | 66.50 | 5.71 | 50.46 | 0.11 | 31.59 |
| | LLaMa-3.2-1B | 33.68 | 30.57 | 46.89 | 91.90 | 29.65 | 94.75 | 36.50 | 51.99 |
| | LLaMa-3.1-8B | 32.84 | 28.69 | 47.94 | 90.91 | 28.20 | 94.09 | 32.35 | 50.72 |
| | MMedS-8B | 33.27 | 31.80 | 47.58 | 91.51 | 28.96 | 94.44 | 32.87 | 51.49 |

## C.2 Benchmarking MIMIC-ICD53

To further evaluate the quality of our constructed dataset MIMIC-ICD53, we employed various unimodal methods to train and test its performance. For the visual modality, we selected ResNet-18, ResNet-50 [22], ViT [15], and Swin Transformer [39], along with two SOTA CXR-specific pretrained models, UniChest [14] and RadDino [45] as the baselines. Based on both computational efficiency and data leakage prevention considerations, we ultimately designated ResNet-50 as the standard visual backbone for the main experiments. ViT is also investigated in ablation studies in Section C.6.

For electronic health record (EHR) data, we first leveraged its inherent tabular structure by treating each EHR attribute as an input dimension, with corresponding numerical values assigned to their respective dimensions. We experimented with MLP and a SOTA tabular data process method TabFPN [24], but both exhibited suboptimal performance. Consequently, we reformatted the EHR data into natural language text using the following template: *"Laboratory values within the 4–7 range indicate normal levels, values 0–3 suggest clinically low levels, and values 8–10 denote elevated levels. The current panel includes [ATTRIBUTE] with the discretized value of [VALUE]..."*. We then evaluated the reformatted EHR data using classic backbone ClinicalBERT [55], BioClinicalBERT [2], and PubMedBERT [18]. We also included natural and medical LLMs LLaMa [54] and MMed-S [58]. Specifically, we replace the last layer of LLMs with a classification head to adapt to the text classification task. Due to the high consumption of computing resources, we only use LoRA [25] to fine-tune the LLM-based models. We further conducted experiments using the same text encoders for the report modality alone and reports combined with EHR data. As shown in Table 6, unimodal performance analysis reveals that the report modality achieves the highest diagnostic results, followed by the EHR modality. Combining the two modalities in text form (Report+EHR) can bring significant performance gains. For the visual modality, the performance gap between different backbone architectures is relatively small. While for the text-based modality, LLM-based

Table 7: Performance with variance on four datasets of different anatomies. All results are calculated over 5 independent runs.

| Dataset | Method | F1 | Precision | Recall | AUC | mAP | Acc | Acc-S | Avg |
|---------|--------|----|-----------|--------|-----|-----|-----|-------|-----|
| MIMIC-ICD53 (Chest) | MedFuse | 34.46±0.0077 | 31.36±0.0082 | 45.04±0.0004 | 90.85±0.0168 | 31.77±0.0084 | 95.34±0.0127 | 41.44±0.0239 | 52.89±0.0085 |
| | BiomedCLIP | 32.99±0.0058 | 29.56±0.0073 | 45.04±0.0007 | 88.71±0.0061 | 29.91±0.0061 | 94.72±0.0032 | 39.83±0.0087 | 51.54±0.0048 |
| | KAD | 36.32±0.0107 | 33.80±0.0125 | 48.33±0.0019 | 91.95±0.0165 | 33.54±0.0111 | 95.12±0.0049 | 40.27±0.0254 | 54.19±0.0104 |
| | DrFuse | 34.10±0.0067 | 33.70±0.0067 | 45.34±0.0244 | 89.50±0.0287 | 31.19±0.0073 | 94.68±0.0639 | 38.25±0.0239 | 52.39±0.0086 |
| | HEALNet | 35.42±0.0075 | 32.76±0.0079 | 47.95±0.0016 | 88.80±0.0137 | 31.97±0.0081 | 94.90±0.0204 | 40.10±0.0209 | 53.13±0.0076 |
| | RAD | **39.71**±0.0101 | **39.07**±0.0099 | **54.74**±0.0016 | **93.00**±0.0103 | **36.74**±0.0116 | **95.40**±0.0050 | **42.33**±0.0228 | **57.28**±0.0089 |
| FairVLMed (Eye) | MedFuse | 81.33±0.0010 | 76.13±0.0021 | 87.29±0.0003 | 87.99±0.0034 | 88.76±0.0049 | 79.50±0.0024 | 79.50±0.0024 | 83.50±0.0020 |
| | BiomedCLIP | 81.27±0.0014 | 72.87±0.0034 | 91.88±0.0005 | 87.69±0.0041 | 87.62±0.0038 | 78.35±0.0044 | 78.35±0.0044 | 83.28±0.0024 |
| | KAD | 81.18±0.0028 | 73.92±0.0080 | 90.03±0.0010 | 88.62±0.0137 | 88.88±0.0158 | 78.65±0.0101 | 78.65±0.0101 | 83.55±0.0064 |
| | DrFuse | 81.69±0.0028 | 73.72±0.0090 | 91.59±0.0022 | 89.33±0.0217 | 90.38±0.0204 | 79.00±0.0121 | 79.00±0.0121 | 84.29±0.0076 |
| | HEALNet | 81.80±0.0011 | 75.22±0.0028 | 89.64±0.0001 | 89.60±0.0030 | 90.45±0.0041 | 79.60±0.0022 | 79.60±0.0022 | 84.39±0.0019 |
| | RAD | **84.30**±0.0028 | **77.52**±0.0070 | **92.38**±0.0005 | **91.32**±0.0126 | **91.88**±0.0144 | **82.40**±0.0080 | **82.40**±0.0080 | **86.63**±0.0060 |
| SkinCAP (Skin) | MedFuse | 79.25±0.0418 | 85.96±0.0538 | 77.99±0.0036 | 96.50±0.0194 | 73.61±0.0363 | 99.34±0.0166 | 74.36±0.0148 | 83.86±0.0223 |
| | BiomedCLIP | 81.49±0.1073 | 87.13±0.1228 | 81.41±0.0091 | 97.22±0.0351 | 79.22±0.1114 | 99.11±0.0282 | 74.36±0.1184 | 85.71±0.0646 |
| | KAD | 82.06±0.1025 | 86.79±0.1290 | 81.27±0.0147 | 97.80±0.0454 | 80.40±0.1066 | 99.25±0.0244 | 75.46±0.1098 | 86.15±0.0654 |
| | DrFuse | 81.18±0.0389 | 85.70±0.0470 | 79.64±0.0040 | 94.92±0.0185 | 76.42±0.0365 | 99.29±0.0158 | 77.66±0.0122 | 84.97±0.0208 |
| | HEALNet | 82.20±0.0890 | 88.69±0.1130 | 81.18±0.0186 | 92.68±0.0225 | 77.97±0.0925 | 99.37±0.0176 | 78.39±0.0480 | 85.79±0.0475 |
| | RAD | **85.48**±0.0678 | **89.48**±0.0750 | **83.23**±0.0136 | **97.97**±0.0356 | **83.55**±0.0639 | **99.48**±0.0159 | **81.32**±0.0474 | **88.64**±0.0407 |
| NACC (Brain) | MedFuse | 31.53±0.0005 | 25.59±0.0001 | 68.36±0.0051 | 85.50±0.0038 | 24.49±0.0004 | 87.44±0.0110 | 58.45±0.0196 | 54.48±0.0011 |
| | BiomedCLIP | 34.36±0.0013 | 29.02±0.0008 | 66.95±0.0002 | 84.00±0.0043 | 26.03±0.0008 | 88.80±0.0010 | 58.21±0.0004 | 55.34±0.0008 |
| | KAD | 35.09±0.0024 | 29.68±0.0039 | 64.49±0.0008 | 85.88±0.0052 | 27.73±0.0026 | 89.69±0.0013 | 57.86±0.0071 | 55.77±0.0028 |
| | DrFuse | 34.11±0.0030 | 27.86±0.0032 | **68.96**±0.0085 | 82.88±0.0070 | 27.88±0.0024 | 87.99±0.0191 | 51.31±0.0045 | 54.43±0.0025 |
| | HEALNet | 35.91±0.0008 | 28.92±0.0004 | 67.33±0.0049 | 85.04±0.0037 | 26.13±0.0006 | 89.55±0.0090 | 56.79±0.0001 | 55.67±0.0008 |
| | RAD | **37.65**±0.0015 | **36.24**±0.0049 | 65.78±0.0003 | **87.11**±0.0019 | **30.03**±0.0023 | **90.36**±0.0010 | **59.64**±0.0078 | **58.12**±0.0020 |

models generally outperform BERT-based models. Among the BERT-based models, ClinicalBERT consistently achieves the best performance. Considering model size and practicality, we selected ClinicalBERT as the default text encoder in our RAD framework.

## C.3 The Variance of Baselines

Due to space limitations, we do not show the variance of the baselines in Table 1. Here we add the variance of all baselines in Table 7. It can be observed that the overall variance of SkinCAP is the largest among all datasets. Meanwhile, there is no significant gap between the variance of different methods, all methods exhibit stable performance across the four datasets.

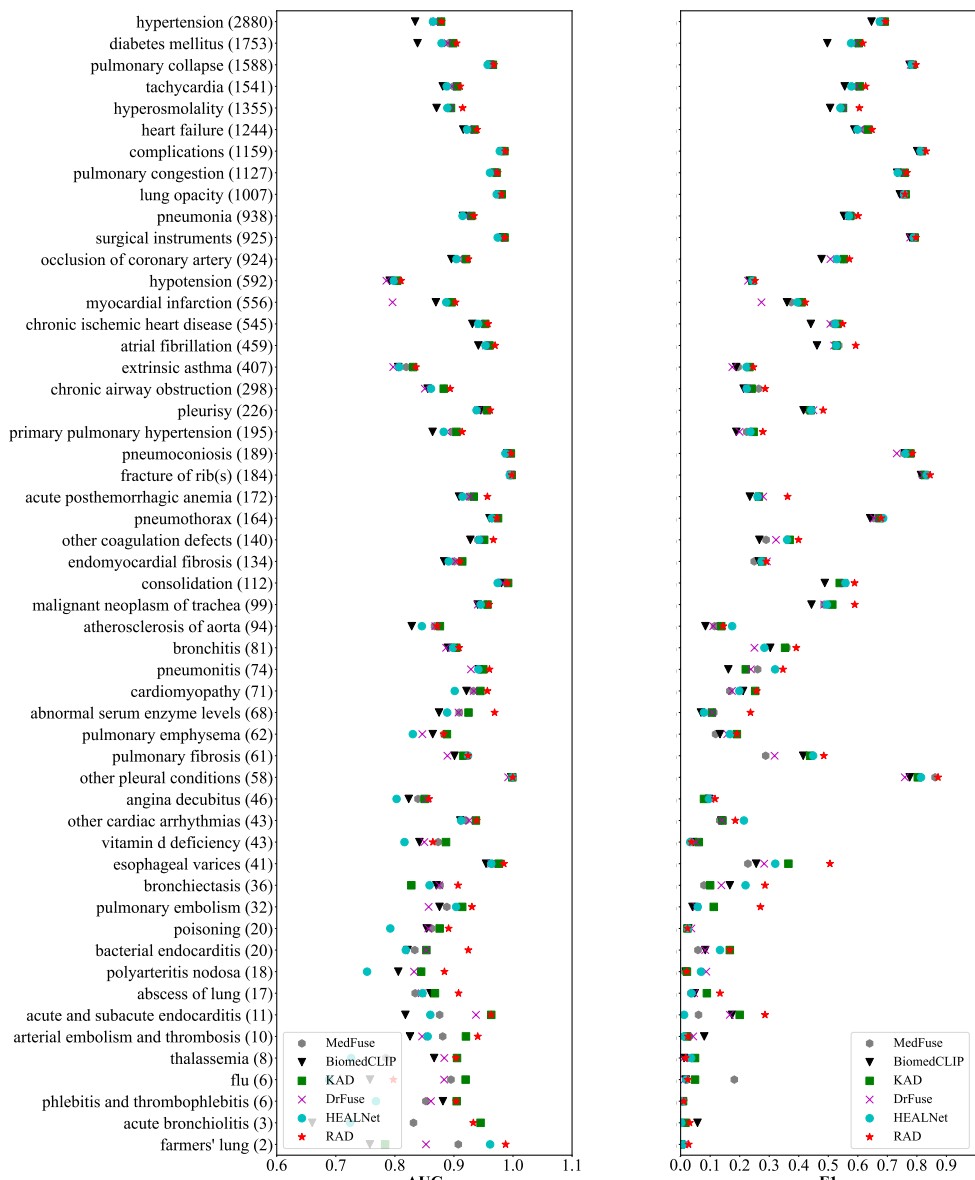

Figure 10: Detailed AUC and F1 for each class in MIMIC-ICD53. The y-axis is the disease name. The numbers in brackets represent the number of samples with this disease.

## C.4   Label-wise Analysis of MIMIC-ICD53

In addition to evaluating the overall performance of RAD in Section 4.2, we also investigated its comprehensive performance across all categories on MIMIC-ICD53. As illustrated in Figure 10, our method achieved the highest scores in 41 out of 53 categories across both AUC and F1 metrics. Furthermore, in the long-tail categories (classes with fewer than 100 positive samples), our method outperformed the previous SOTA by 1.60% in AUC and by 4.44% in F1. Importantly, the performance gains in these long-tail categories exceeded the average improvements observed across all categories, underscoring the robustness and practical utility of RAD under real-world clinical settings.

## C.5   Interpretability Cases

In this subsection, we further explore the textual interpretability of RAD by presenting additional visualization cases. In addition, the full names of the abbreviations in Table 4 are given here. In Table 4, the "PC" is short for Platelet Count, "Bilirubin" is Serum Bilirubin, "ALT" is Alanine

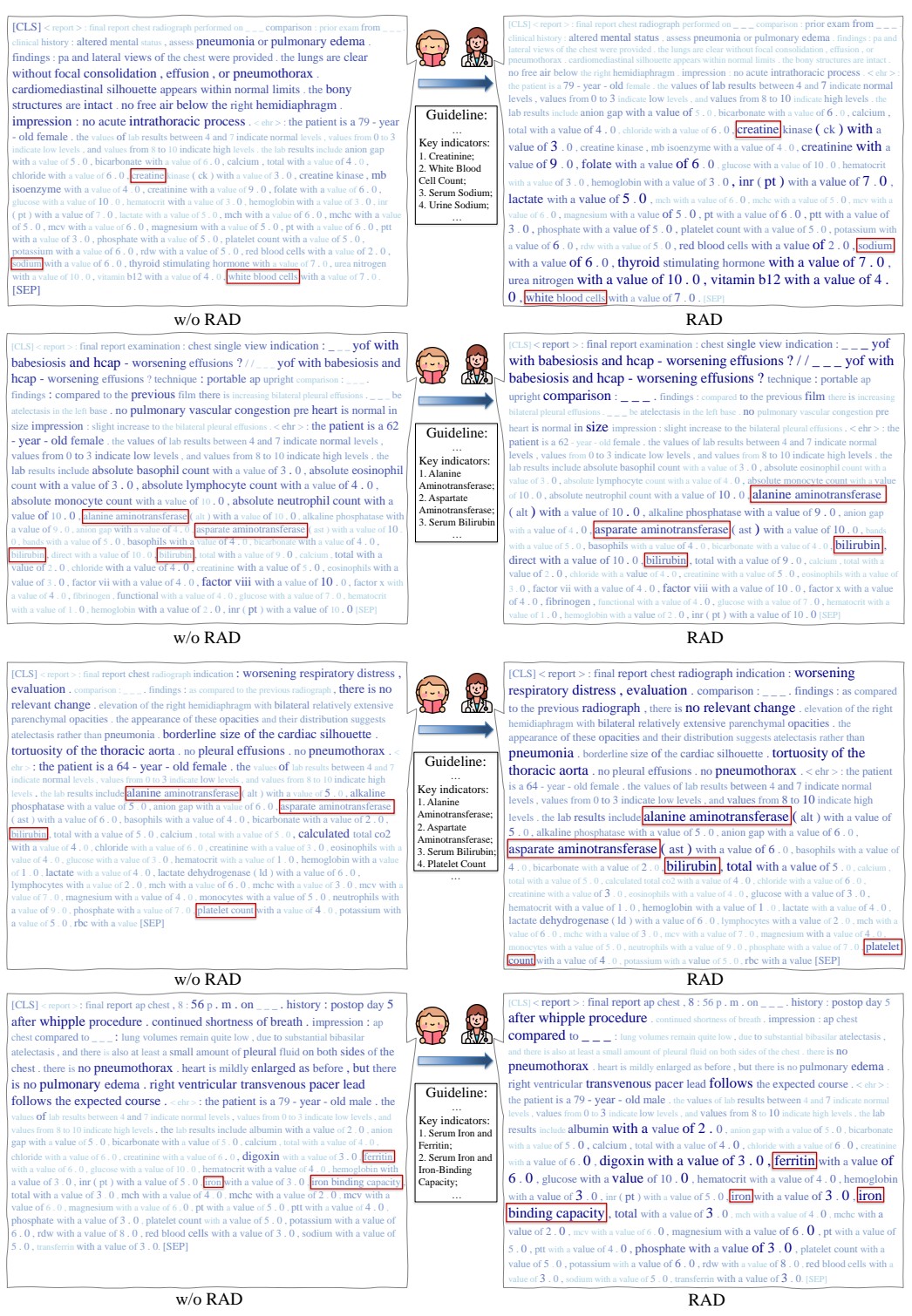

Figure 11: Visualization of model attention to textual content. Both font size and color intensity reflect attention magnitude, with red highlighting disease-critical indicators mentioned in the guideline.

Aminotransferase, "IBC" is Iron-Binding Capacity, "WBC" is White Blood Cell Count, and "AST" is Aspartate Aminotransferase. Figure 11 presents longer and clearer cases of interpretability on the textual data. The third row is the complete content of Figure 1, and the other rows are other cases

Table 8: Performance across different combinations of encoder backbones on MIMIC-ICD53. Subscript with arrows represents the absolute improvement. Our method is highlighted with shading.

| Backbone | Method | F1 | Precision | Recall | AUC | mAP | Acc | Acc-S | Avg |
|---|---|---|---|---|---|---|---|---|---|
| ResNet+ ClinicalBERT | w/o RAD | 34.91 | 31.01 | 50.91 | 91.27 | 32.24 | 94.50 | 38.63 | 53.35 |
| | RAD | $39.71_{4.80\uparrow}$ | $39.07_{8.06\uparrow}$ | $54.74_{3.83\uparrow}$ | $93.00_{1.73\uparrow}$ | $36.74_{4.50\uparrow}$ | $95.40_{0.90\uparrow}$ | $42.33_{3.70\uparrow}$ | $57.28_{3.93\uparrow}$ |
| ViT+ ClinicalBERT | w/o RAD | 37.22 | 35.77 | 45.64 | 91.01 | 34.05 | 95.63 | 41.04 | 54.34 |
| | RAD | $41.21_{3.99\uparrow}$ | $41.14_{5.37\uparrow}$ | $51.89_{6.25\uparrow}$ | $92.70_{1.69\uparrow}$ | $37.24_{3.19\uparrow}$ | $95.78_{0.15\uparrow}$ | $41.97_{0.93\uparrow}$ | $57.42_{3.08\uparrow}$ |
| ResNet+ BioClinicalBERT | w/o RAD | 36.71 | 33.76 | 49.99 | 92.03 | 34.31 | 95.02 | 38.86 | 54.38 |
| | RAD | $39.95_{3.24\uparrow}$ | $39.90_{6.14\uparrow}$ | $51.72_{1.73\uparrow}$ | $92.59_{0.56\uparrow}$ | $36.34_{2.03\uparrow}$ | $95.89_{0.87\uparrow}$ | $42.29_{3.43\uparrow}$ | $56.95_{2.57\uparrow}$ |
| ViT+ BioClinicalBERT | w/o RAD | 36.32 | 34.35 | 48.99 | 92.08 | 33.39 | 95.18 | 39.77 | 54.30 |
| | RAD | $40.00_{3.68\uparrow}$ | $39.58_{5.23\uparrow}$ | $50.70_{1.71\uparrow}$ | $92.25_{0.17\uparrow}$ | $36.38_{2.99\uparrow}$ | $96.01_{0.83\uparrow}$ | $42.52_{2.75\uparrow}$ | $56.78_{2.48\uparrow}$ |

Table 9: Ablation on LLM refinement of RAD on the MIMIC-ICD53.

| LLM-refine | F1 | Precision | Recall | AUC | mAP | Acc | Acc-S | Avg |
|---|---|---|---|---|---|---|---|---|
| ✗ | 38.73 | 36.94 | 53.24 | 92.99 | 36.56 | 95.34 | 40.43 | 56.32 |
| ✓ | 39.71 | 39.07 | 54.74 | 93.00 | 36.74 | 95.40 | 42.33 | 57.28 |

Table 10: Ablation on retrieval knowledge sources. "Ours" is the default setting with four knowledge sources. "+ Google Search" means adding a new source based on "Ours". "- Random Drop" means randomly removing one knowledge source for each guideline.

| | Source | F1 | Precision | Recall | AUC | mAP | Acc | Acc-S | Avg |
|---|---|---|---|---|---|---|---|---|---|
| Single Source | Wiki | 39.77 | 39.11 | 47.05 | 93.14 | 36.93 | 96.02 | 41.67 | 56.24 |
| | Research | 38.54 | 36.35 | 51.41 | 93.01 | 36.26 | 95.47 | 40.69 | 55.96 |
| | Guideline | 39.79 | 39.17 | 50.32 | 93.03 | 37.12 | 96.02 | 41.42 | 56.70 |
| | Book | 39.49 | 39.14 | 47.65 | 93.11 | 36.84 | 96.20 | 42.04 | 56.35 |
| Multi Source | - Random Drop | 40.18 | 40.15 | 49.34 | 93.05 | 37.35 | 96.24 | 43.32 | 57.09 |
| | Ours | 39.71 | 39.07 | 54.74 | 93.00 | 36.74 | 95.40 | 42.33 | 57.28 |
| | + Google Search | 40.56 | 40.01 | 50.56 | 92.84 | 36.97 | 96.15 | 42.89 | 57.14 |

with different indicators. It can be observed that RAD enables the model to dynamically focus on indicators valuable for the current diagnostic goal based on the retrieved guidelines.

## C.6 Ablation Study

In this part, we conduct a comprehensive ablation study to systematically evaluate the impact of architectural backbones, key components, and hyperparameter configurations in RAD.

**Ablation on different backbones.** In Section 4.4, we demonstrated the impact of RAD on model performance when replacing different modality backbones, as reflected in the average metrics, AUC, F1, and mAP. To provide a more comprehensive evaluation, we have included additional metrics in Table 8, such as Precision, Recall, Accuracy, and Acc-S, which collectively illustrate the holistic enhancement of the model in diagnostic tasks.

**Ablation on LLM refinement of the retrieved knowledge.** To assess the necessity of LLM refinement in Section 3.2.1, we further conducted an ablation study by comparing RAD with and without this step. Specifically, we constructed baseline guidelines through direct concatenation of top-$k$ retrieved documents and evaluated the performance on MIMIC-ICD53. The results in Table 9 demonstrate that all metrics have decreased after removing the LLM filtering step, underscoring the importance of regularizing the retrieved text. The LLM refinement not only performs semantic filtering to eliminate irrelevant contexts but also standardizes heterogeneous medical knowledge into actionable diagnostic guidelines—a critical enabler for effective downstream knowledge infusion.

**Ablation on Knowledge Sources.** To investigate the effect of modifying the knowledge base on model performance, we compare each knowledge source's individual performance, as well as the performance of adding or removing sources based on our default setting. As presented in Table 10, clinical guidelines provide the most valuable knowledge, as they directly encode established

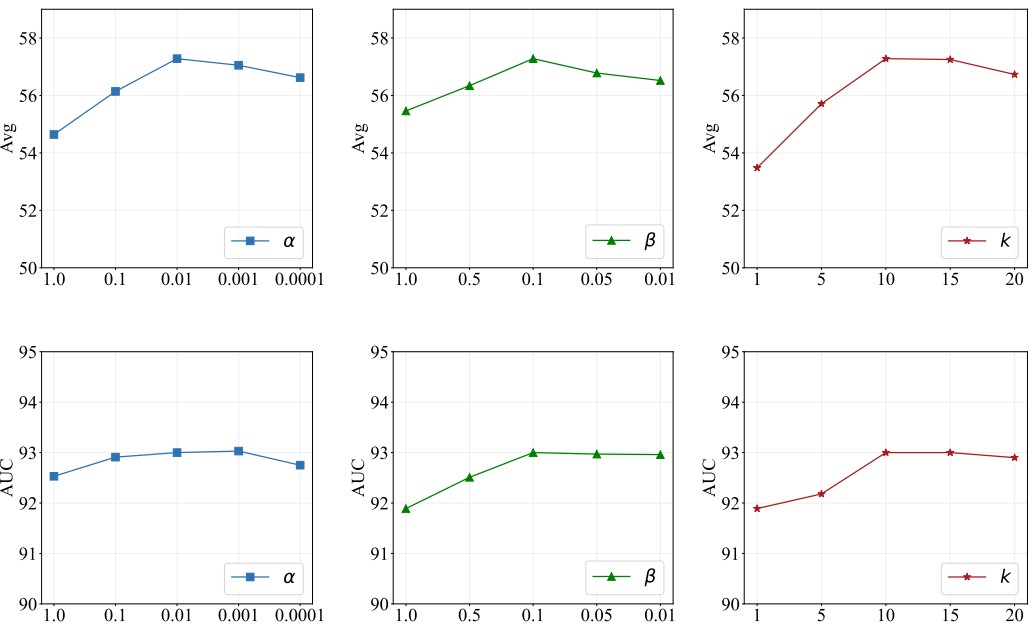

Figure 12: Analysis of hyper-parameters on MIMIC-ICD53.

diagnostic criteria, key indicators, and decision pathways specifically designed for clinical practice. Research papers show the lowest contribution, as they often focus on novel discoveries, experimental treatments, or specialized cases rather than established diagnostic standards. For well-established diseases, diagnostic criteria have become a consensus, making cutting-edge research less useful. When applying multiple knowledge sources, the performance of RAD remains stable across different source counts (±0.2 Avg), demonstrating RAD's robustness to knowledge base modifications.

## C.7  Cost Analysis.

To evaluate the practical feasibility of our framework, we analyze the additional cost of RAD brought by the guideline acquisition process. Since we only perform retrieval at the label level, which avoids the prohibitive cost of sample-wise retrieval. The retrieval process incurs negligible computational overhead. The additional cost primarily occurs during the LLM refinement phase, where the retrieved documents are processed by LLMs for each label of the dataset. When using Qwen2.5-72B model, the average processing time is 33.83s per label. The total preprocessing time for guideline retrieval and refinement on MIMIC-ICD53 is around 31 minutes. The cost can be further reduced using smaller LLMs. When expanding to new datasets, the linear growth of retrieval cost $\mathcal{O}(N_{disease})$ ensures efficient scalability, as it grows significantly slower than patient samples $\mathcal{O}(N_{sample})$ in real-world scenarios. Furthermore, the retrieval and refinement steps are executed once per dataset during preprocessing, eliminating runtime delays during clinical deployment. In general, RAD achieves knowledge infusion with minimal practical overhead.

## C.8  Hyper-parameter analysis

To evaluate the impact of hyperparameters in RAD, we conduct experimental analysis on the three key hyperparameters $\alpha$, $\beta$, and top-$k$. The hyperparameter $\alpha$ determines the weight of the guideline-enhanced contrastive learning for visual and text features. And $\beta$ determines the weight of binary cross-entropy loss and the guideline-enhanced contrastive loss. Top-$k$ controls the number of retrieved documents for each disease (label). Figure 12 presents the performance trends as these parameters vary. As $\beta$ decreases, the model performance initially improves before declining. This pattern arises because an excessively high weight over-prioritizes the auxiliary loss, disrupting the optimization of the primary classification loss. On the contrary, a very low weight also leads to performance degradation, underscoring the utility of the guideline in refining multi-modal feature representations. $\alpha$ exhibits a similar pattern. The optimal values for $\alpha$ and $\beta$ are $1e-2$ and $1e-1$,

respectively. Regarding the top-$k$ hyperparameter, the model achieves worst performance at $k = 1$, with gradual improvement as $k$ increases. However, performance plateaus after reaching a threshold ($k = 10$ here). When retrieving too few documents, limited informative content leads to suboptimal results. Conversely, retaining excessive documents beyond the threshold primarily introduces noisy knowledge, as core disease-related information has already been captured within the top-ranked documents.

