# OpenReview forum: "RAD: Towards Trustworthy Retrieval-Augmented Multi-modal Clinical Diagnosis"
_NeurIPS.cc/2025/Conference — NeurIPS 2025 poster_

### Official Review · Reviewer_TyCK · 2025-06-29

**Clarity:** 3
**Significance:** 2
**Originality:** 2
**Rating:** 5
**Confidence:** 4

**Summary:**

The paper presents a model for clinical diagnosis based on images and medical records, guided by "knowledge" about a set of specific diseases. The overall model consists of visual and text encoders that feed corresponding embeddings into attention-based label and guideline decoders. The model is trained and evaluated on several datasets demonstrating improved performance versus other models across all datasets. The paper appears to be comprehensive, with relevant ablation studies, derivation of loss function, details of datasets, etc., however some model training details are omitted, such as the number of epochs.

**Questions:**

It would be helpful to understand which of the four sources used for knowledge retrieval are the most relevant to the performance of the model. For example, would just the clinical practice guidelines be sufficient?

Do you think the results can be improved if a more balanced dataset is used? Or can some data augmentation be used to enrich the parts of the dataset where only a few samples are present?

**Ethical Concerns:**

["NO or VERY MINOR ethics concerns only"]

**Final Justification:**

The authors provided sufficient new details that clarify my initial concerns.

**Limitations:**

The only limitation listed by the authors is due to the static retrieval knowledge corpus. I would not consider this to be a limitation as new knowledge can always be added to the knowledge base and the model can be re-trained. I think a bigger limitation of this work is a limited number of classes of diseases the model is trained to recognize as this is a classifier model. New diseases cannot be recognized even if there is an adequate knowledge about them is present in the knowledge base.

**Paper Formatting Concerns:**

Paper format is OK.

**Quality:**

3

**Strengths And Weaknesses:**

Overall quality of the work seems good, with good justification, detailed enough description of the proposed model and datasets, ablation studies, etc. Some additional details and experiments would strengthen the work, see "Questions" section of this review. While the work demonstrates clear improvements compared to other models, these improvements are still not sufficient to make this work practical as a medical diagnostics tool.

While not entirely novel, model guidance through augmentation of relevant knowledge retrieved from four different sources seems to be helpful to achieve the stated performance. This is definitely a strength of this work.

Nice to see the construction of another dataset, MIMIC-ICD53. However this dataset seems to be hugely disbalanced. Not surprising, Figure 8 indicates model's low performance for many diseases that do not have sufficient number of samples.

---

> ### Author Rebuttal · Authors · 2025-07-31
>
> We sincerely thank Reviewer TyCK for your valuable time and constructive feedback. We appreciate your recognition of our paper quality and the construction of the MIMIC-ICD53 dataset. In the following, we provide our responses point-by-point.
>
> >Weakness 1 and Limitation: Improvements are still not sufficient to make this work practical as a medical diagnostics tool. And a bigger limitation of this work is a limited number of classes of diseases the model is trained to recognize as this is a classifier model. New diseases cannot be recognized even if there is an adequate knowledge about them is present in the knowledge base.
>
> **New diseases**:
>
> We would like to explain that a key advantage of RAD's architecture is its inherent zero-shot potential through the text-query mechanism:
>
> Unlike standard classifiers (e.g., MLP) with fixed output shapes, RAD uses text queries (both disease labels and guidelines) as inputs to the decoder. This means the model can flexibly process any disease name, regardless of whether it was seen during training. Given a new disease, we can directly retrieve the corresponding guideline with the disease name, and then perform zero-shot classification at the lowest cost.
> Following your question, we conducted zero-shot experiments on the Chest-Det dataset (which contains 10 new disease categories). We use checkpoints trained on the MIMIC-ICD53 dataset to perform zero-shot inference.
>
> Chest-Det|Acc|AUC|mAP|F1|Avg
> -|-|-|-|-|-
> BiomedCLIP|38.93|56.73|22.57|31.39|37.41
> KAD|40.33|53.91|21.59|30.19|36.51
> Ours|54.63|60.26|24.47|32.16|42.88
>
> RAD demonstrates meaningful zero-shot capability, achieving 60.26 AUC on unseen diseases.
> And our method significantly outperforms baselines, proving that RAD can effectively leverage the knowledge base for novel diseases. For medical AI, even modest accuracy improvements can significantly reduce diagnostic errors when scaled across thousands of cases.
>
> **Practical Utility as a Diagnostics tool**:
>
> We acknowledge that the current absolute performance is still not enough for full clinical application. However, we believe RAD offers meaningful advancement in its current form:
>
> 1. The RAD model is **sufficient to work as a medical diagnostic tool on common diseases**:
>
>     Diseases|AUC|mAP|Acc|F1|Precision|Recall|Avg
>     -|-|-|-|-|-|-|-
>     Heart disease, cardiomegaly|98.68|89.61|82.5|75.5|90.94|95.69|88.82
>     Pulmonary collapse|96.38|84.44|79.38|77.85|80.98|93.56|85.43
>     Lung opacity|98.12|83.28|76.66|66.4|90.67|94.64|84.96
>
>     The above presents RAD's performance on some common chest diseases. The model already achieves performance comparable to senior clinicians, making it suitable as a diagnostic tool for these diseases.
>
> 2. **Interpretability**: Unlike other baselines, RAD can perform evidence-based diagnosis by following clinical guidelines. RAD's knowledge-grounded approach provides transparent diagnosis process that clinicians can evaluate and trust - a critical factor for real-world adoption that pure performance metrics don't capture.
>
> 3. **Adaptability**: As shown, the framework can incorporate new knowledge for zero-shot inference, surpassing existing baselines. Furthermore, as a small discriminative model, RAD can achieve strong performance on new diseases at the lowest cost by simple retraining.
>
> These discussions can better elucidate RAD's real-world practicality, and we will include them in our revised manuscript as a ''Discussion'' section. We sincerely appreciate your insightful questions and suggestions!
>
> >Question 1: It would be helpful to understand which of the four sources used for knowledge retrieval are the most relevant to the performance of the model. For example, would just the clinical practice guidelines be sufficient?
>
> To address this, we conducted ablation studies where we used only one knowledge source at a time:
>
> Source|F1|Precision|Recall|AUC|mAP|Acc|Acc-S|Avg
> -|-|-|-|-|-|-|-|-
> Wiki|39.77|39.11|47.05|93.14|36.93|96.02|41.67|56.24
> Research|38.54| 36.35| 51.41|93.01|36.26|95.47|40.69|55.96
> Guideline|39.79| 39.17|50.32|93.03|37.12|96.02|41.42|56.70
> Book|39.49|39.14|47.65|93.11|36.84|96.20|42.04|56.35
> All|39.71|39.07|54.74|93.00|36.74|95.40|42.33|57.28
>
> Results on MIMIC-ICD53 demonstrate:
> 1. **Clinical guidelines provide the most valuable knowledge** (Avg: 56.70), as they directly encode established diagnostic criteria, key indicators, and decision pathways specifically designed for clinical practice. This source aligns most with the gold standard for diagnosing specific diseases.
> 2. **Research papers show the lowest contribution** (Avg: 55.96), as they often focus on novel discoveries, experimental treatments, or specialized cases rather than established diagnostic standards. For well-established diseases, diagnostic criteria have become consensus, making cutting-edge research less useful.
> 3. **Multiple knowledge sources are complementary**: While guidelines alone already provide near-optimal results, the complete knowledge base (4 sources) can further boost the performance.
>
> For resource/knowledge-constrained settings, clinical guidelines alone are nearly sufficient for maintaining high accuracy. However, for best practice, the full knowledge base is recommended.
>
> This important ablation will be included in Section 4.4 of our revised manuscript.
>
> >Question 2: Do you think the results can be improved if a more balanced dataset is used? Or can some data augmentation be used to enrich the parts of the dataset where only a few samples are present?
>
> Yes, we agree that class imbalance greatly affects performance on tail classes. If a more balanced dataset and data augmentation methods are used, the performance should be improved. In the following, we have conducted experiments as suggested:
> - Resample: Oversampling 10 tail classes with a sampling ratio of 5
> - Resample+Mixup: Further applying a popular data augmentation method Mixup[1] on tail classes based on the resampled dataset
>
> All strategies are applied on the training set, and we use mixup with α=0.2 (suggested in the paper). The performance on tail classes is reported:
>
> Dataset|F1|Precision|Recall|AUC|mAP|Avg
> -|-|-|-|-|-|-
> Original|5.53|4.07|45.42|91.41|1.32|29.55
> Resample|9.18|10.33|35.85|91.31|3.20|29.97
> Resample+Mixup|9.98|14.06|32.97|91.27|3.71|30.40
>
> Although the average performance shows little increase, F1, Precision, and mAP exhibit significant improvement. After resampling, mAP has increased by more than 100% (1.32 to 3.2). Mixup augmentation provided additional gains beyond simple resampling, particularly in precision.
>
> Since RAD is **orthogonal** to data-balancing methods, these methods can be easily integrated into RAD to further improve the performance on tail classes.
>
> Additionally, we would like to clarify **why choosing an imbalanced dataset**. In healthcare settings, disease distributions naturally follow long-tail patterns (e.g., pneumonia is much more common than farmers' lung/pneumoconiosis). Thus, we intentionally preserved the original distribution of MIMIC to align with real-world medical settings.
>
> The tail classes resampled here are: 'acute bronchiolitis and unspecified acute lower respiratory infection', 'farmers' lung and pneumoconiosis due to asbestos and mineral fibers', 'thalassemia', 'acute and subacute endocarditis', 'phlebitis and thrombophlebitis', 'flu due to avian flu virus', 'arterial embolism and thrombosis', 'acute and subacute bacterial endocarditis', 'polyarteritis nodosa', 'abscess of lung and mediastinum, pyothorax, mediastinitis'.
>
> [1]mixup: Beyond empirical risk minimization (ICLR 2018)

---

> > ### Comment · Reviewer_TyCK · 2025-08-05
> >
> > Thank you for addressing my concerns, I am generally satisfied with these additional details.  I think this work is quite acceptable and I will change my rating.

---

> > > ### Author Response · Authors · 2025-08-05
> > >
> > > Thank you very much for your recognition and support of our work. We will carefully incorporate the reviewers' suggestions in the subsequent version.

---

### Official Review · Reviewer_jbMM · 2025-07-01

**Clarity:** 3
**Significance:** 3
**Originality:** 3
**Rating:** 5
**Confidence:** 4

**Summary:**

This work introduces a framework (RAD) that explicitly injects disease-specific clinical guidelines into multimodal diagnostic models. The pipeline (i) retrieves disease-centric documents from four curated medical corpora and refines them with a large-language-model summariser, (ii) aligns image and text representations to these guidelines via a Guideline-Enhanced Contrastive Loss (GECL) term, and (iii) fuses modalities with a dual‐decoder transformer whose queries are the guideline text and the disease labels. A dual-axis evaluation scheme is proposed to quantify interpretability. RAD is evaluated against other competitors across four anatomies (chest, fundus, dermatology, and brain).

**Questions:**

- Figure 2 depicting the Dual Diagnostic Network appears asymmetric. The two branches seem to contain different amounts of information, with the guidance branch being inclusive/indicative of the label. It is unclear if this dual-branch design is strictly necessary, and an ablation study on a single-branch design would strengthen the argument for its necessity (though current study swaps to an MLP, not one-branch).

- The training processes for other comparison methods are not clearly elaborated, making it difficult to ascertain the fairness of the comparison with existing methods. If RAD's improvements are primarily due to learning more external knowledge, which other methods might not have been exposed to, then the observed gains might stem largely from this additional knowledge rather than the pipeline novelty itself. If so, can RAG's knowledge injection strategy/framework be applied to other models—for example, BiomedCLIP?

- Equation (7) is an important loss of RAD and is the base of the method. But it's not clear to me if the the text encoder, which is used to obtain embeddings for input text and guidelines, is optimized?  If the encoders are frozen, the GECL term can only shift the lightweight projection heads or prototypes, so its ability to “pull” patient text T toward guideline G seems limited.  Conversely, if the encoders are updated, both T and G are re-encoded each step. In this case, how to ensure the guideline embedding G still faithfully represent the original guideline semantics?


- Providing a couple of complete examples of the crafted condition guidelines in the main paper (or a prominent appendix section) would greatly enhance reader understanding of what these guidelines look like and could help extend the research. The appendix shows the prompt template, not concrete examples.

- While Figure 1 and Table 3 show that RAD pays more attention to representative laboratory indicators/keywords. The current evidence—higher guideline-recall scores and attention heat-maps—mainly shows correlation: GECL already pulls representations toward guideline tokens, so a larger recall is almost guaranteed even if the model’s final decision does not truly hinge on those tokens. To demonstrate that RAD’s predictions are causally driven by guideline evidence, consider adding a counter-factual test (e.g., mask or perturb key indicators at inference) to see if performance drops.

**Ethical Concerns:**

["NO or VERY MINOR ethics concerns only"]

**Final Justification:**

The submission was a solid work but I had a few detailed technical questions about the method. The rebuttal addressed all my concerns and I support the acceptance of this submission.

**Limitations:**

Yes.

**Paper Formatting Concerns:**

None.

**Quality:**

3

**Strengths And Weaknesses:**

Strengths
- Overall, the motivation behind this work is clear and aligns with current calls for trustworthy medical AI, where expert knowledge is critical to ensuring reliable decision-making. The work is well-written and the method is clearly explained.

- The experimental setup is robust, and the RAD method's applicability is successfully demonstrated across diverse medical modalities (chest, eye, skin, and brain). The extensive evaluations on four datasets validate its generalizability and superiority against state-of-the-art baselines.

- The proposed Guideline-enhanced Feature Constraint (GECL) is empirically proven to be an effective component, contributing to the improved performance and interpretability of RAD.

Weaknesses
- It remains unclear that the dual-decoder design—one of the paper’s headline contributions—provides value beyond a simpler alternative.

- Because training protocols for baselines are under-specified, the performance gap may partly reflect unequal access to external knowledge rather than genuine architectural superiority, casting doubt on fair comparison.

- The optimisation scope of GECL, which is an important component of the proposed method, is ambiguous, making it hard to judge whether the claimed guideline-driven reasoning is robust or merely correlational.

(Details are in the questions)

---

> ### Author Rebuttal · Authors · 2025-07-31
>
> We sincerely thank Reviewer jbMM for your valuable time and constructive feedback. We appreciate your recognition of our clear motivation and robust experimental validation. Below, we provide our responses to each point.
>
> >Weakness & Question 1: Necessity of the dual-decoder design.
>
> Thank you for the suggestion. To address your concern, we have conducted an ablation study comparing four different decoder structures:
> - MLP: A single MLP head
> - Label-branch: Only the label decoder branch
> - Guideline-branch: Only the guideline decoder branch
> - Dual-branches: Our dual-decoder structure
>
> The experimental results on MIMIC-ICD53 are as follows:
>
> Model|F1|Precision|Recall|AUC|mAP|Acc|Acc-S|Avg
> -|-|-|-|-|-|-|-|-
> MLP|39.34|37.74|51.87|92.94|36.36|95.59|39.95|56.26
> Label-branch|40.00|39.78|50.80|92.59|36.35|95.88|42.29|56.81
> Guideline-branch|39.65|38.96|53.60|92.76|36.43|95.42|41.37|56.88
> Dual-branches|39.71|39.07|54.74|93.00|36.74|95.40|42.33|57.28
>
> Key findings:
>
> 1. The dual-branch design achieves the best average performance. Neither single branch alone can match the performance of the dual-branch architecture (while still better than MLP).
>
> 2. While the guideline branch's query contains label information, the two branches can still learn complementary information.
>
> Analysis: The guideline branch achieves higher recall (53.60% vs 50.80%), which means it tends to predict positive cases more aggressively. While the label branch gives more conservative predictions, showing higher precision (39.78% vs 38.96%) and sample-wise accuracy (42.29% vs 41.37%). This is because the guideline branch can make bold diagnoses based on the symptoms/key indicators mentioned in the guideline, minimizing false negatives. While the label branch (with only the disease names as the query) develops stricter decision boundaries. Thus, **combining the two branches can further improve performance**.
>
> We will include this ablation study in Section 4.4 of our revised manuscript.
>
> >Weakness & Question 2: Training protocols. If RAD's improvements are primarily due to learning more external knowledge, which other methods might not have, the observed gains might stem largely from this additional knowledge rather than the pipeline novelty itself. If so, can RAD's injection strategy be applied to other models—for example, BiomedCLIP?
>
> We apologize for not clearly detailing the training protocols. To ensure fair comparison, all models were trained under identical conditions, with the same optimizer (AdamW,lr=1e-4, weight_decay=0.01), training epochs, and so on. Detailed configuration for each dataset is available in the config files of our anonymized code repository.
>
> To address your concern about knowledge access and application to other models, we conducted the following experiments:
> - BiomedCLIP+Guideline: Adding the guideline (same as RAD) into the input of BiomedCLIP
> - BiomedCLIP+RAD: Adding RAD's knowledge injection strategy to BiomedCLIP
>
> Dataset|Model|F1|Precision|Recall|AUC|mAP|Acc|Acc-S|Avg
> -|-|-|-|-|-|-|-|-|-
> -|BiomedCLIP|32.99|29.56|45.04|88.71|29.91|94.72|39.83|51.54
> ICD53|BiomedCLIP+Guideline|32.57|30.68|46.89|90.05|29.55|94.36|38.18|51.75
> -|BiomedCLIP+RAD|34.99|33.11|48.94|91.70|32.40|94.86|39.89|53.70
> -|BiomedCLIP|81.49|87.13|81.41|97.22|79.22|99.11|74.36|85.71
> Skin|BiomedCLIP+Guideline|81.60|86.95|79.49|97.62|79.72|99.38|78.20|86.14
> -|BiomedCLIP+RAD|84.70|88.87|82.66|98.00|82.48|99.44|80.59|88.11
>
> Results on two datasets show that BiomedCLIP+Guideline can only bring a slight improvement over the original model. While combining RAD on BiomedCLIP can markedly enhance performance (>2% Avg). This demonstrates that **RAD's gains are not solely from the external knowledge**. Its superiority stems from our novel pipeline design rather than unequal knowledge access. The pipeline is essential for effective knowledge integration.
>
> >Weakness & Question 3: The optimisation scope of GECL. It's not clear to me if the text encoder, is optimized? If the encoders are frozen, the GECL term can only shift the lightweight projection heads or prototypes, so its ability to “pull” patient text T toward guideline G seems limited. Conversely, if the encoders are updated, both T and G are re-encoded each step. How to ensure the guideline embedding G still faithfully represent the original guideline semantics?
>
> Thank you for raising this insightful suggestion! In our original implementation, all components, including the text encoder are optimized. We agree that updating the text encoder may affect the faithfulness of G. Our implementation will be changed to the frozen version. We conducted experiments to investigate how training the text encoder influences the performance.
>
> Method|F1|Precision|Recall|AUC|mAP|Acc|Acc-S|Avg
> -|-|-|-|-|-|-|-|-
> Tuneable|39.71|39.07|54.74|93.00|36.74|95.40|42.33|57.28
> Frozen|39.65|38.96|54.60|93.00|36.73|95.42|42.35|57.24
>
> Results show that the performance difference was minimal (only -0.04% Avg).
>
> We think that lightweight projection heads have sufficient capacity to align patient text with guideline semantics without modifying the fundamental text representations. This is because the module operates on the embedding learned by the pretrained text encoder, which is sufficiently compact, semantical, and easily transformed through lightweight projector layers.
>
> Moreover, the contrastive learning objective in GECL primarily adjusts the relative positions in the embedding space rather than radically changing individual embeddings. This maintains the semantic integrity of guidelines while allowing for task-specific alignment.
>
> We will revise this implementation detail of the frozen text encoder in future updates.
>
> Regarding your concern about whether RAD  is "robust or merely correlational," please refer to our response to Question 5.
>
> >Question 4: Examples of the crafted guidelines.
>
> Below is an example for the guideline of "bronchitis":
>
> "### Summary of Key Diagnostic Features for Bronchitis
>
> **Disease Description:**\nBronchitis is an inflammation of the bronchi, the air passages in the lungs. It can be classified into two main types: acute and chronic. Acute bronchitis is typically a self-limiting condition characterized by a cough that may produce sputum and is often caused by viral infections. Chronic bronchitis, on the other hand, is a long-term condition defined by a productive cough lasting for at least three months in two consecutive years, often associated with chronic obstructive pulmonary disease (COPD). The primary risk factor for chronic bronchitis is tobacco smoking, with other factors including air pollution and occupational exposures.
>
> **Important Lab Tests and Values:**\n- Acute Bronchitis:\n  White Blood Cell Count (WBC): Usually normal or slightly elevated.\n  C-reactive Protein (CRP): May be slightly elevated but not typically high.\nSputum Culture: Not routinely necessary, but can be useful if bacterial infection is suspected. Chronic Bronchitis:\n  Pulmonary Function Tests (PFTs): Reduced FEV1/FVC ratio, indicating airflow obstruction.\nSputum Analysis: Increased mucus production, often with neutrophil infiltration.\nBlood Gas Analysis:** May show hypoxemia and hypercapnia in advanced cases.
>
> **Key Radiological or Clinical Findings:**\nAcute Bronchitis:\n Chest X-ray: Usually normal, but may show hyperinflation or peribronchial thickening.\nPhysical Examination: Wheezing, crackles, and rhonchi on auscultation. Chronic Bronchitis:\n Chest X-ray: May show hyperinflation, increased bronchovascular markings, and signs of emphysema.\nCT Scan: Can reveal bronchial wall thickening and mucus plugging.\n Physical Examination: Barrel chest, cyanosis, and signs of cor pulmonale in advanced cases.
>
> **Diagnostic Symptoms or Relevant Clinical Features:**\nAcute Bronchitis:\n Cough: Initially dry, then becomes productive with clear or yellowish sputum.\n Fever: Usually mild or absent; high fever suggests pneumonia.\n Fatigue and Body Aches: Common but generally mild.\n Wheezing and Shortness of Breath: May be present, especially in patients with underlying asthma.
> Chronic Bronchitis:\n Cough: Persistent, productive cough with sputum, often for at least three months in two consecutive years.\n Dyspnea: Shortness of breath, especially on exertion.\n  Wheezing: Common, especially in the morning.\n  Chest Pain: May occur due to prolonged coughing.\n  Fatigue and Malaise: Persistent, often due to chronic hypoxemia."
>
> More examples of full guidelines will be added to Appendix B.2.
>
> >Question 5: Counter-factual test of causality
>
> Thank you for this insightful suggestion. We conducted a counter-factual test by randomly masking key indicators in the input text at inference time. We masked 0%, 10%, 30%, and 50% of the indicators of the MIMIC-ICD53 test set.
>
> mask|F1|Precision|Recall|AUC|mAP|Acc|Acc-S|Avg
> -|-|-|-|-|-|-|-|-
> 0%|39.71|39.07|54.74|93.00|36.74|95.40|42.33|57.28
> 10%|38.44|35.80|50.61|92.76|35.54|95.55|39.34|55.43
> 30%|34.74|31.60|49.53|91.84|32.26|94.75|37.97|53.24
> 50%|29.74|25.17|47.14|89.91|26.31|93.85|34.03|49.45
>
> At lower masking ratios (10%), the model exhibits relatively small performance degradation. However, as the masking ratio increases, we observe a significant performance drop (10% F1 decrease at 50% masking). This demonstrates that model performance is highly sensitive to the presence of key indicators. (The model can still maintain certain performance even with 50% masking because RAD has multimodal inputs, including image, report, and EHR.)
>
> The results confirm that **RAD's predictions are causally dependent on guideline-relevant information**, not merely correlational. Our model's decision-making process hinges on these specific clinical indicators identified by the guideline.
>
> We will include this part in our revised manuscript, providing more evidence that RAD follows the guideline to focus on key indicators.

---

> > ### Comment · Reviewer_jbMM · 2025-08-02
> >
> > I thank the authors for spending time to carefully address my concerns through additional experiments. My questions have been all answered satisfactorily, and I am raising my score to 5.

---

> > > ### Author Response · Authors · 2025-08-02
> > > **Reply Rebuttal Comment by Authors**
> > >
> > > Thank you very much for your recognition and support of our work. We will carefully incorporate the reviewers' suggestions in the subsequent version.

---

### Official Review · Reviewer_UQga · 2025-07-02

**Clarity:** 2
**Significance:** 2
**Originality:** 2
**Rating:** 3
**Confidence:** 4

**Summary:**

This paper introduces RAD, a Retrieval-Augmented Diagnosis framework that enhances multimodal clinical diagnosis by injecting refined disease-specific external knowledge (e.g., clinical guidelines, medical textbooks, research papers) into the diagnostic process. RAD uses a three-pronged approach: (1) retrieving and refining guideline knowledge, (2) embedding this knowledge in multimodal representations via a contrastive loss, and (3) integrating it through a dual diagnostic decoder. The authors also propose a dual-axis interpretability evaluation framework.

**Questions:**

1. How does RAD handle errors or outdated content in the knowledge corpus or LLM-generated summaries? Have you analyzed the effect of injecting incorrect or incomplete guidelines on downstream accuracy or interpretability?

2. Attention weights are used to proxy adherence to diagnostic indicators. Have you considered augmenting your interpretability evaluation with external expert review or ground-truth decision rationales?

3. Could you expand on the differences between your method and retrieval-augmented generation or multimodal RAG approaches?

**Ethical Concerns:**

["NO or VERY MINOR ethics concerns only"]

**Final Justification:**

The authors have satisfactorily addressed most of my concerns, so I am increasing my rating accordingly.

**Limitations:**

yes

**Quality:**

2

**Strengths And Weaknesses:**

Strengths
- RAD introduces a systematic method to inject disease-specific guidelines into the entire diagnostic pipeline, including input augmentation, feature extraction, and decision fusion, rather than limiting knowledge use to pretraining stages.
- Across four anatomically diverse datasets (MIMIC-ICD53, FairVLMed, SkinCAP, NACC), RAD consistently outperforms strong baselines like BiomedCLIP and KAD in terms of F1, AUC, and sample-wise accuracy.


Weaknesses
1. The main novelty lies in the explicit combination and systematic integration of prior methods (retrieval, LLM summarization, contrastive alignment, and dual decoder); however, the technical advances in each are modest individually. There is limited comparison or differentiation from conventional retrieval-augmented generation or discriminative frameworks. The distinction from RAG approaches is only briefly justified.
2. The approach assumes that a suitable disease guideline exists, that LLM summaries are reliable, and that retrieved content truly covers all relevant diagnostic edge cases. The refinement of retrieved documents via Qwen2.5-72B is central to RAD, yet the quality and consistency of LLM-generated summaries are not critically evaluated.
3. While improved averages are shown, the paper does not analyze or visualize errors where adherence to retrieved guidelines leads to mistakes or misses patient-specific context.
4. While interpretability evaluation is a valuable addition, the reliance on attention weights (text and image) as evidence of adherence is controversial.
5. The pipeline for retrieving, refining, and updating guidelines is resource-intensive (uses Qwen2.5-72B), potentially limiting reproducibility in resource-constrained or privacy-sensitive settings.
6. The paper benchmarks against baselines in multimodal medical AI, but omits RAG-style methods tailored to clinical reasoning or explanation. Explanation for specific baseline choices or additional comparisons would strengthen the empirical claim.
7. Occasional jargon (“dual diagnostic transformer decoder”, “feature prototypes”), ambiguous notation in equations (Section 3.2.2), and dense referencing to appendices can break the narrative flow.
8. While the manuscript emphasizes clinical workflows, an analysis of clinician utility, end-user validation, or prospective trials in realistic deployment environments is missing.

---

> ### Author Rebuttal · Authors · 2025-07-31
>
> We sincerely thank Reviewer UQga for your valuable time and constructive feedback. Below, we provide our responses to each point.
>
> >W1 & Q3: The main novelty lies in the explicit combination and systematic integration of prior methods (retrieval, LLM summarization, contrastive alignment, and dual decoder); technical advances in each are modest individually. Limited comparison from conventional RAG/multimodal RAG/discriminative frameworks.
>
> We sincerely apologize for any confusion caused to the reviewer, and would like to clarify the novelty of RAD, which is not merely a combination of existing techniques.
>
> - First, we acknowledge that retrieval and LLM summarization are not our pioneering contributions. Our retrieval goal is simple: obtaining disease-specific knowledge. This purpose is sufficiently addressed by standard retrieval with the disease name as the query. Disease names already provide precise semantics for retrieving relevant knowledge. We did not propose novel query optimization or search strategy, since existing technology is enough to meet our needs.
>
> - Second, RAD is a novel method for effectively injecting guidelines into the diagnosis model. Specifically, we propose utilizing guidelines as anchors and design the GECL loss to simultaneously align text and images, aiming to reduce the distance between multimodal representations and corresponding guidelines. To further utilize guidelines to steer the fusion of multimodal information, we design a dual-decoder structure to explicitly leverage guidelines for cross-modal interaction. **The GECL loss and the dual-decoder are our novel design tailored for medical knowledge injection**.
>
> - Finally, another major contribution of RAD lies in advancing trustworthy medical AI by designing **a comprehensive evaluation framework and metrics to measure the model's interpretability**, which doctors favour.
>
> In the following, we enumerate the major differences between RAD and other methods:
> Method|RAD|RAG|Multimodal RAG|Discriminative Baselines
> -|-|-|-|-
> Task Type|Discriminative|Generative (QA)|Generative (VQA)|Discriminative
> Task-specific Knowledge|√|√|√|×
> Retrieval Goal|Diagnosic Guidelines|Dynamic Knowledge|Similar Cases|-
> Retrieval Granularity|Label-level|Sample-level|Sample-level|-
> Retrieval Difficulty|Easy|Hard|Hard|-
> Injection Approach|Input augmentation (text), feature learning, and decision-making|Input augmentation (text)|Input augmentation (text/image)|-
> Technical Focus|Inject knowledge to guide training|Search strategy, query optimization...|Retriever optimization, Reranking...|Pretraining, multimodal fusion
>
> For RAG & multimodal RAG, the fundamental difference lies in the goal of retrieval. RAG methods usually dynamically retrieve textual knowledge for current questions; Multimodal RAG primarily retrieves similar data samples(e.g., image-report pairs), especially for medical multimodal RAG[1]. While our method retrieves diagnostic standards once to constrain training on all samples. And we incorporate the knowledge to guide multimodal representation learning, modality fusion, participating in entire training pipeline. Our method also has major differences with RAG in retrieval difficulty, retrieval complexity, etc.
>
> Discriminative methods typically do not inject domain-specific knowledge on downstream tasks and often focus solely on pre-training on extensive corpus for text encoders. While our approach internalizes guidelines not only within text encoders but also impacts all parameters, including visual encoders and dual decoders.
>
> We will revise Section 2 (Related Work) to better explain these distinctions.
>
> [1] MMed-RAG: Versatile Multimodal RAG System for Medical Vision Language Models
>
> >W2: The quality and consistency of LLM-generated summaries are not critically evaluated.
>
> Thank you for the suggestion. In fact, all final guideline versions underwent rigorous verification by junior doctors from a major hospital. Three critical aspects are checked by doctors, including: MIMIC-ICD53 construction process, final guideline correctness, and double-checking the indicators for Guideline Recall calculation. Specifically, for MIMIC-MM53, a total of 53 guidelines were reviewed, and physicians made moderate modifications to only 6 of them.
>
> We further compare the performance with/without manual verification and refinement via a smaller LLM:
> Model|F1|Precision|Recall|AUC|mAP|Acc|Acc-S|Avg
> -|-|-|-|-|-|-|-|-
> Qwen2.5-72B w/ manual|39.71|39.07|54.74|93.00|36.74|95.40|42.33|57.28
> Qwen2.5-72B w/o manual|40.24|40.15|50.23|92.72|37.01|96.17|43.33|57.12
> Qwen2.5-7B w/o manual|40.20|39.89|49.94|93.16|37.16|96.09|42.92|57.05
>
> Even without manual check, the model only shows a negligible performance drop. This is because the LLM only needs to perform simple text summarization tasks rather than generating novel content.
>
> >W3 & Q1: Analyze errors where adherence to guidelines leads to mistakes or misses patient-specific context. The effect of injecting incorrect/incomplete guidelines on downstream accuracy
>
> Thank you for the suggestion. Please refer to experiments and discussion in response to Reviewer QEAp's Weakness 5.
>
> >W4 & Q2: The reliance on attention weights (text and image) as evidence is controversial. Have you considered augmenting your interpretability evaluation with external expert review or ground-truth decision rationales?
>
> Although attention-based methods have some limitations, we should note that
>
> 1. Attention weight-based methods like Grad-CAM[1] have been widely accepted as standard interpretability tools[2,3]. In medical tasks, attention-based localization is widely used, with validation against physician-annotated ground-truth[4,5].
> 2. Our visual interpretability metrics in Table 4 directly align with physician-verified lesion annotations (which are the gold standard in diagnosis, enough to serve as ground-truth decision rationales). This high performance provides clinical validation beyond pure attention visualization.
> 3. In clinical practice, medical guidelines represent gold standards that have become common consensus. Only highly recognized diagnostic criteria can be included in textbooks and guidelines. The RAD model can exhibit high-level alignment with such guidelines, demonstrating that its interpretability is relatively reliable.
>
> [1]Grad-cam: Visual explanations from deep networks via gradient-based localization
>
> [2]Transformer interpretability beyond attention visualization
>
> [3]Attention-based Interpretability with Concept Transformers
>
> [4]A foundation model for generalizable disease detection from retinal images
>
> [5]Large-scale pancreatic cancer detection via non-contrast CT and deep learning
>
> >W5: Retrieving, refining guidelines is resource-intensive (uses Qwen2.5-72B), limiting reproducibility in resource-constrained/privacy-sensitive settings.
>
> As detailed in Appendix C.7 of our submission, the computational overhead of retrieving is minimal due to our offline label-level retrieval strategy. The cost scales with O(N_disease) rather than O(N_sample), making it highly efficient. The LLM refinement takes 33.83 seconds per disease label. Total preprocessing time takes only 31 minutes for the largest dataset. More importantly, these operations are one-time preprocessing steps that introduce no runtime delays during model training/deployment. And using smaller LLMs (e.g., Qwen2.5-7B) can further reduce this cost with minimal performance drop (result table in our response to Weakness 2).
>
> For privacy-sensitive settings, we use the LLM to summarize the retrieved knowledge for certain diseases, not the patient context. Thus, the LLM usage will not disclose patients' privacy.
>
> >W6: Omits RAG-style methods. Explanation for specific baseline choices or additional comparisons would strengthen the claim.
>
> To address your concern, we implemented RAG baselines on MLLMs like Qwen2.5-VL. The experiments are on the single-label dataset FairVLMed. Specifically, we use the same guidelines used by our model as the RAG input for fair comparison.
> Method|Model|F1|Precision|Recall|Acc|Avg
> -|-|-|-|-|-|-
> w/o RAG|Qwen2.5-VL-7B|57.46|73.88|47.01|63.60|60.49
> w/ RAG|Qwen2.5-VL-7B|59.68|75.00|49.56|65.55|62.45
> w/o RAG|HuatuoGPT-Vision-7B|65.19|61.39|69.50|62.05|64.53
> w/ RAG|HuatuoGPT-Vision-7B|66.66|58.59|77.32|60.45|65.76
> w/o RAG|Lingshu-7B|28.91|73.60|17.98|64.85|46.34
> w/ RAG|Lingshu-7B|29.90|77.20|18.54|64.95|47.65
> -|Ours|84.30|77.52|92.38|82.40|84.15
>
> Even with RAG, the performance gaps between MLLMs and discriminative models are still significant. This indicates that adding the guideline in input of generative models does not offer significant improvement, since diagnosis requires fundamentally different knowledge integration than generative tasks.
>
> >W7: Jargon (“dual diagnostic transformer decoder”), ambiguous notation (Section 3.2.2), and dense referencing to appendices break the narrative flow.
>
> We sincerely apologize for the confusion. The manuscript will be revised to replace jargon with clear descriptions (e.g., from "dual diagnostic transformer decoder" to "dual decoder"). For a smooth narrative flow, we will put all references to appendix at the end of each section. We will also revise Section 3.2.2 to provide clearer notations.
>
> >W8: Analysis of clinician utility, end-user validation, or prospective trials in realistic deployment environments
>
> Thank you for your valuable suggestion. As a primary method exploration, our ultimate goal is to promote clinical practice. We are currently collaborating with the thoracic surgery department at a major hospital to deploy RAD for auxiliary examination. However, due to patient privacy protection and the need to fulfill relevant procedures, the ethics approval process is still ongoing. We will include the results and analysis of clinical validation once all regulatory procedures are done. But we believe that, for NeurIPS, the current extensive validation on open datasets is still persuasive and fits this community.

---

> > ### Comment · Reviewer_UQga · 2025-08-06
> >
> > Thank you for addressing most of my concerns, and I will raise my rating.

---

> > > ### Author Response · Authors · 2025-08-07
> > >
> > > Thank you very much for your suggestion and support of increasing the score. We will carefully consider your suggestions and incorporate them in the subsequent version.

---

> ### Author Response · Authors · 2025-08-04
>
> Dear Reviewer UQga,
>
> Thanks again for your valuable time and thoughtful feedback. We have tried our best to address your concerns in our rebuttal.
>
> If any points remain unclear or require further clarification, we would be glad to provide more details.
>
> Best,
>
> The authors of Submission 5515

---

### Official Review · Reviewer_QEAp · 2025-07-07

**Clarity:** 3
**Significance:** 3
**Originality:** 2
**Rating:** 4
**Confidence:** 5

**Summary:**

This paper introduces Retrieval-Augmented Diagnosis (RAD), a framework designed to enhance credible multimodal clinical diagnosis. Current AI-driven medical research often relies on implicitly encoded knowledge, ignoring the needs of specific tasks. RAD solves this problem by explicitly injecting external medical knowledge through the following three key mechanisms: retrieving and optimizing disease-centric guidelines, applying a guideline-enhanced contrastive loss function to align features, and using a dual diagnosis decoder to guide cross-modal fusion. The paper also constructs the MIMIC-ICD53 dataset and conducts experiments on 4 datasets to verify the effect. The paper has sufficient work, but may need to be strengthened in theory.

**Questions:**

1.Introduce the innovation of the retrieval process and give an explanation from a theoretical level.

2.Analyze cases where retrieved reports are incorrect.

3.72B Offline models may introduce bias and should be supplemented with manual verification instructions or ablations of different large model refinements.

4.The comparative experiments lack comparison results with large language models, such as Qwen-vl, llava-med, and huatuogpt-vision.

**Ethical Concerns:**

["NO or VERY MINOR ethics concerns only"]

**Final Justification:**

The authors have mostly addressed my questions.

**Limitations:**

yes

**Quality:**

3

**Strengths And Weaknesses:**

Strengths

1. This paper systematically integrates multi-source medical knowledge into the diagnostic pipeline, addressing the limitation of implicit knowledge encoding in prior methods.

2. The MIMIC-ICD53 dataset is constructed to provide a reliable testing benchmark.

3. Extensive experiments are conducted on four datasets to effectively verify the model.

Weaknesses

1. No contribution in search strategy. Search strategy is the most important part of RAD, but this article is not innovative enough in search method and directly uses MedCPT as a search engine.

2. Retrieval mainly focuses on text modality and lacks the utilization of multimodal information.

3. The LLM-based knowledge refinement and dual decoder structure may introduce higher computational costs, though the authors note retrieval is label-level and manageable.

4. The framework relies on a fixed knowledge base and lacks experimental effects of modifying the knowledge base or discussions on knowledge base updating algorithms.

5. There is a lack of discussion on the correctness of the retrieved results. If the wrong report is retrieved, it may lead to further error accumulation.

6. The class labels of MIMIC-ICD53 are too unbalanced, which may bring serious obstacles to model training.

7. The data refinement process directly uses the 72B offline model and lacks manual verification, which may lead to data deviation.

---

> ### Author Rebuttal · Authors · 2025-07-31
>
> We sincerely thank Reviewer QEAp for your valuable time and constructive feedback. Below, we provide our responses to each point.
>
> >Weakness & Question 1: No contribution in search strategy/retrieval process.
>
> We apologize for some potential misunderstanding, and would like to clarify that our **research focuses on how to effectively inject retrieved knowledge into multi-modal discriminative models, rather than designing novel search algorithms**. Specifically, the novelty of RAD lies in the effective injection of disease-specific diagnostic knowledge acquired by the widely used retriever MedCPT[1,2] with the disease name as the query.
>
> Although search strategy innovation is important for dynamic retrieval in RAG systems, it is not necessary for our setting. Our retrieval goal is simply obtaining disease-specific knowledge. This purpose is sufficiently addressed by standard retrieval with MedCPT, since disease names already provide precise semantics for guideline retrieval.
>
> We will revise Section 3.2 of our paper to clarify this research focus and reduce any potential misunderstandings about our contribution.
>
> [1] Benchmarking retrieval-augmented generation for medicine
>
> [2] Improving retrieval-augmented generation in medicine with iterative follow-up questions
>
> >Weakness 2: Retrieval lacks the utilization of multimodal information.
>
> Essentially, aligning the machine diagnostics with the clinical guidelines makes the model more trustworthy. This is the target of injecting the clinical guideline knowledge into the model learning instead of multimodal information, which is beyond our scope. Diagnostic guidelines—by their nature—are primarily textual knowledge that defines standardized diagnostic criteria.
>
> For our framework, medical knowledge injection requires the knowledge sources to be authoritative and standardized (e.g., "The levels of amyloid and tau proteins in cerebrospinal fluid are valuable for the diagnosis of Alzheimer's disease"). This differs fundamentally from case-based retrieval in multimodal RAG, which seeks similar patient examples. To clearly differentiate our method from standard RAG and multimodal RAG approaches, we have included a comparative analysis table, to which we refer the reviewer in our response to Reviewer UQga's Weakness 1.
>
> >Weakness 3: The LLM-based refinement and dual decoder may introduce higher computational costs.
>
> **LLM-refinement**:
>
> As detailed in Appendix C.7 of our submission, the computational overhead of retrieving is minimal due to our offline label-level retrieval strategy. The cost scales with O(N_disease) rather than O(N_sample), making it highly efficient. The LLM refinement takes 33.83 seconds per disease label. Total preprocessing time takes only 31 minutes for the largest dataset MIMIC-ICD53. More importantly, these operations are one-time preprocessing steps that introduce no runtime delays during model training/deployment. And using smaller LLMs (e.g., Qwen2.5-7B) can further reduce this cost with minimal performance drop (result table in our response to Weakness 7).
>
> **Dual-Decoder**:
>
> The computational impact of our dual-decoder is negligible, as shown by component-level analysis:
>
> Component|Params|FLOPs
> -|-|-
> Vision Encoder|31.33M (16%)|87.82G (80.13%)
> Text Encoder|135.92M (71%)|21.77G (19.86%)
> Dual Decoder|25.22M (13%)|12.65M (0.01%)
>
> Despite accounting for 13% of parameters, the dual decoder contributes only 0.01% of total FLOPs. Because the decoder structure is simply a transformer layer that operates on already-processed embeddings (not raw inputs).
>
> This confirms that both components introduce small computational overhead while delivering considerable improvements.
>
> >Weakness 4: Experimental effects of modifying the knowledge base or discussions on knowledge base updating algorithms.
>
> Thank you for the suggestion. In the following, we conducted experiments that modify the number of knowledge sources to evaluate RAD's robustness:
> - Default: Original version with 4 sources
> - \- Random Drop: For each disease, randomly remove one knowledge source for guideline retrieval
> - \+ Google Search: Adding "Google Search" as a new retrieval source
>
> Sources|F1|Precision|Recall|AUC|mAP|Acc|Acc-S|Avg
> -|-|-|-|-|-|-|-|-
> Default|39.71|39.07|54.74|93.00|36.74|95.40|42.33|57.28
> \- Random Drop|40.18|40.15|49.34|93.05|37.35|96.24|43.32|57.09
> \+ Google Search|40.56|40.01|50.56|92.84|36.97|96.15|42.89|57.14
>
> The performance of RAD remains stable across different source counts (±0.2 Avg), demonstrating RAD's robustness to knowledge base modifications. As shown in our response to the Question 1 of Reviewer TyCK, even single-source implementations (clinical guidelines alone) achieve strong performance (56.70 Avg), significantly outperforming baselines.
>
> Additionally, in clinical practice, medical guidelines represent **gold standards** that have become common consensus. Such knowledge evolves slowly (especially for most common diseases), making frequent knowledge base updates unnecessary for most scenarios.
>
> >Weakness 5 & Question 2: Lack of discussion on the correctness of the retrieved results. If the wrong report is retrieved, it may lead to further error accumulation.
>
> To address this, we conducted controlled experiments where we intentionally replaced 10% and 20% of retrieved guidelines with irrelevant disease guidelines during training:
> Wrong%|F1|Precision|Recall|AUC|mAP|Acc|Acc-S|Avg
> -|-|-|-|-|-|-|-|-
> 0|39.71|39.07|54.74|93.00|36.74|95.40|42.33|57.28
> 10%|39.91|39.56|49.30|92.94|36.94|95.95|41.97|56.65
> 20%|38.66|37.42|49.20|92.89|36.24|95.59|40.09|55.73
> Label-branch|40.00|39.78|50.80|92.59|36.35|95.88|42.29|56.81
>
> As shown in the above table, the*wrong knowledge only leads to little performance degradation, demonstrating RAD's built-in error resilience. This **robustness stems from our dual-decoder architecture**: The label branch operates independently using only disease labels as query. Using the label branch alone can achieve 56.8 average performance. Thus, even when the retrieved guidelines are wrong, RAD can still provide robust results.
>
> This architecture can further prevent error accumulation by prioritizing the label decoder when guideline information is unreliable, maintaining performance around 56.8 Avg.
>
> >Weakness 6: The unbalanced labels of MIMIC-ICD53.
>
> First, the class imbalance indeed brings obstacles. However, we would like to clarify **why choosing an imbalanced dataset**. In healthcare settings, disease distributions naturally follow long-tail patterns (e.g., pneumonia is much more common than farmers' lung/pneumoconiosis). Thus, we intentionally preserved the original distribution of MIMIC to align with real-world medical settings.
>
> Besides, we also conducted an experiment on a balanced dataset to demonstrate the effectiveness of RAD. Please kindly refer to our response to Reviewer TyCK's Question 2.
>
> >Weakness 7 & Question 3: The data refinement process lacks manual verification. 72B models should be supplemented with manual verification instructions or ablations of different large model refinements
>
> We apologize for not clearly detailing our validation process. All final guideline versions underwent rigorous verification by junior doctors from a major hospital. In fact, three critical aspects are checked by physicians, including: MIMIC-ICD53 construction process, final guideline correctness, and double-checking the indicators for Guideline Recall calculation.
>
> Furthermore, we compare the performance with/without manual verification and summarization using a smaller model:
> Model|F1|Precision|Recall|AUC|mAP|Acc|Acc-S|Avg
> -|-|-|-|-|-|-|-|-
> Qwen2.5-72B w/ manual|39.71|39.07|54.74|93.00|36.74|95.40|42.33|57.28
> Qwen2.5-72B w/o manual|40.24|40.15|50.23|92.72|37.01|96.17|43.33|57.12
> Qwen2.5-7B w/o manual|40.20|39.89|49.94|93.16|37.16|96.09|42.92|57.05
>
> Even without manual check, the model only shows a negligible performance drop. This is because the LLM only needs to perform simple summarization tasks on retrieved knowledge rather than generating novel content. And the minimal performance drop between 7B and 72B model demonstrates that 7B model is enough for the summarization task. We will use smaller models in future implementations to lower the costs.
>
> >Question 4: Experiments lack comparison results with large language models, such as Qwen-vl, llava-med, and huatuogpt-vision.
>
> We have conducted comprehensive comparisons with state-of-the-art multimodal large language models (MLLMs) by converting the data into VQA format. The results are on the FairVLMed dataset since it is the only single-label dataset suitable for VQA. For each MLLM, we evaluated both few-shot and supervised fine-tuning (SFT) settings:
>
> Model|Setting|F1|Precision|Recall|Acc|Avg
> -|-|-|-|-|-|-
> Qwen2.5-VL-7B|Few-shot|57.46|73.88|47.01|63.60|60.49
> -|SFT|67.70|72.33|63.63|68.95|68.15
> HuatuoGPT-Vision-7B|Few-shot|65.19|61.39|69.50|62.05|64.53
> -|SFT|69.74|71.64|67.93|69.85|69.79
> Lingshu-7B|Few-shot|28.91|73.60|17.98|64.85|46.34
> -|SFT|71.91|65.93|79.08|68.40|71.33
> Ours|-|84.30|77.52|92.38|82.40|84.15
>
> Significant performance gaps exist between RAD and all MLLMs (>12 Avg), even though the single-label dataset is already the simplest for MLLMs. (Multi-label datasets require the model to complete classifications for all labels at once.)
>
> From the results, we can find that complex diagnostic tasks are better suited for discriminative models than generative MLLMs, as evidenced by the substantial performance gap. Moreover, RAD (and all our baselines) can achieve superior performance with significantly lower resource cost than MLLMs.
>
> Regarding llava-med that the reviewer mentioned, we did not include it as it's outdated (2023) with poor capability. Instead, we add Lingshu-7B, a new medical MLLM released in June 2025. We will add more clarification in our model selection.

---

> > ### Comment · Reviewer_QEAp · 2025-08-02
> > **Response**
> >
> > I thank the authors for their response. The authors have mostly addressed my questions.

---

> > > ### Author Response · Authors · 2025-08-02
> > > **Reply Rebuttal Comment by Authors**
> > >
> > > Thank you very much for your recognition. If you still have any concerns remaining, please inform us and we will try our best to address any points that are not clear during this reviewer-author discussion phase. We will carefully incorporate your suggestions into the final revision, and sincerely appreciate it if you can raise the score to support our substantial rebuttal.

---

### Comment · Area_Chair_4JY1 · 2025-08-04
**Less than 3 days remaining**

This is a reminder that the author-reviewer discussion period will end soon. Please take time to read the other reviews and engage further in the discussion.

The authors have put significant effort into their response, and continued discussion is important for ensuring a thorough and high-quality review process.

---

### Author Response · Authors · 2025-08-09
**Rebuttal and Discussion Summary**

We sincerely thank all reviewers for their insightful feedback and engagement throughout the discussion period. Special thanks to the Area Chair for facilitating this valuable discussion process. We appreciate that the reviewers recognized key strengths of our work:

- Systematic integration of clinical knowledge to advance trustworthy medical AI (Reviewer QEAp, UQga & jbMM)
- Robust experimental validation across diverse medical modalities (Reviewer QEAp, UQga & jbMM)
- Construction of a reliable new dataset MIMIC-ICD53 (Reviewer QEAp & TyCK)
- Clear motivation and well-explained methodology (Reviewer jbMM & TyCK)

During the rebuttal and discussion phases, we have resolved all major concerns through further experiments and clarifications, as explicitly confirmed by all reviewers. Specifically, we:
- Provided critical clarifications:
   - Distinguish RAD from RAG/Multimodal RAG methods through various differences (QEAp W1, W2; UQga W1)
   - Clarify the novelty of RAD (UQga W1)
   - Explained the GECL optimization scope (jbMM Q3)
   - Physician-verified guidelines (QEAp W7; UQga W8; jbMM Q4)
- Conducted extra experiments:
   - Ablations on different modules (QEAp W4, W7; UQga W2; jbMM W1, W3; TyCK Q1)
   - Robustness test (QEAp W5; UQga W3; jbMM Q5)
   - Costs analysis and cost-reducing strategies (QEAp W3; UQga W5)
   - Compilability with baselines and other methods (jbMM W2; TyCK Q2)

All suggestions will be carefully incorporated to improve the quality of our work. We once again express our gratitude to all reviewers for their time and effort devoted to evaluating our work.

---

### Decision · Program_Chairs · 2025-09-17

**Decision:**

Accept (poster)

**Comment:**

This paper proposes Retrieval-Augmented Diagnosis (RAD), a framework that integrates disease-specific clinical knowledge into multimodal diagnostic models through guideline retrieval and refinement, a guideline-enhanced contrastive loss, and a dual-decoder design. The strengths of the paper include a clear motivation rooted in trustworthy medical AI, systematic knowledge injection across the diagnostic pipeline, extensive evaluations on four diverse datasets with consistent improvements over strong baselines and multimodal LLMs, and thorough ablations clarifying the roles of different components. The rebuttal further strengthened the work by demonstrating zero-shot capability, robustness to noisy retrieval, ablations on knowledge sources (with guidelines shown to be most critical), fairness of baseline comparisons, and causal dependence on guideline features.

Remaining weaknesses include limited novelty in retrieval strategy, dependence on textual guidelines, potential resource constraints, controversial reliance on attention-based interpretability, dataset imbalance, and absence of prospective clinical validation. Nonetheless, the reviewers converged on acceptance after rebuttal, agreeing that the methodological contribution, empirical validation, and interpretability advances justify inclusion at NeurIPS.